# Dimer-specific immunoprecipitation of active caspase-2 identifies TRAF proteins as novel activators

Alexander C Robeson[†], Kelly R Lindblom[†], Jeffrey Wojton, Sally Kornbluth[*] & Kenkyo Matsuura[**] iD

## Abstract

Caspase-2 has been shown to initiate apoptotic cell death in response to specific intracellular stressors such as DNA damage. However, the molecular mechanisms immediately upstream of its activation are still poorly understood. We combined a caspase-2 bimolecular fluorescence complementation (BiFC) system with fluorophore-specific immunoprecipitation to isolate and study the active caspase-2 dimer and its interactome. Using this technique, we found that tumor necrosis factor receptor-associated factor 2 (TRAF2), as well as TRAF1 and 3, directly binds to the active caspase-2 dimer. TRAF2 in particular is necessary for caspase-2 activation in response to apoptotic cell death stimuli. Furthermore, we found that dimerized caspase-2 is ubiquitylated in a TRAF2-dependent manner at K15, K152, and K153, which in turn stabilizes the active caspase-2 dimer complex, promotes its association with an insoluble cellular fraction, and enhances its activity to fully commit the cell to apoptosis. Together, these data indicate that TRAF2 positively regulates caspase-2 activation and consequent cell death by driving its activation through dimer-stabilizing ubiquitylation.

**Keywords** apoptosis; Caspase-2; dimer purification; TRAF; ubiquitylation
**Subject Categories** Autophagy & Cell Death
**The EMBO Journal (2018) 37: e97072**

## Introduction

Protein–protein interactions often result in multimeric complexes that induce signaling to significantly alter cellular biology. As such, these protein complexes are tightly controlled by a number of factors such as protein expression or post-translational modifications like phosphorylation or ubiquitylation. Elucidating the biochemistry of these complexes can provide critical insights into biological and pathological developments. However, this research is often limited by the availability of quality tools specific to the known components of the complexes.

One example is the cysteine protease caspase-2. Caspase-2 was one of the earliest caspases to be identified and has increasingly been established as an important modulator of the cellular response to insults such as genotoxic (Sidi *et al*, 2008; Ho *et al*, 2009; Olsson *et al*, 2009; Shalini *et al*, 2016), metabolic (Nutt *et al*, 2005, 2009; Johnson *et al*, 2013; Yang *et al*, 2015), and cytoskeletal stress (Ho *et al*, 2008). Furthermore, its emerging role as a tumor suppressor (Ho *et al*, 2009; Dorstyn *et al*, 2012; Manzl *et al*, 2012; Parsons *et al*, 2013; Puccini *et al*, 2013; Terry *et al*, 2015; Shalini *et al*, 2016) and a promoter of metabolic disorders (Johnson *et al*, 2013; Machado *et al*, 2015, 2016) indicate that modulating its activity could have therapeutic potential. However, the molecular regulation of caspase-2 activation has remained largely obscure.

Apoptotic caspases can be divided into two camps: initiator caspases that begin the apoptotic signaling cascade and executioner caspases that propagate this signal. Caspase-2 is generally classified as an initiator caspase; like some other initiator caspases, caspase-2 contains a long amino-terminal prodomain with a caspase recruitment domain (CARD), and its activation initially occurs through dimerization (Butt *et al*, 1998; Baliga *et al*, 2004). The CARD, which is thought to form a docking site for an activation platform, is critical for this dimerization (Duan & Dixit, 1997; Butt *et al*, 1998; Chou *et al*, 1998; Colussi *et al*, 1998b). The originally described caspase-2 activation platform, the PIDDosome, consisted of PIDD and RAIDD; upon activation, these proteins were found to recruit caspase-2 into high molecular weight signaling aggregates (Tinel & Tschopp, 2004; Park *et al*, 2007). However, the importance of the PIDDosome has been called into question by subsequent studies which have shown that caspase-2 activation can occur independently of either PIDD or RAIDD or both (Manzl *et al*, 2009, 2012; Imre *et al*, 2012; Ribe *et al*, 2012; Peintner *et al*, 2015).

A lack of precise methods limited the study of caspase-2 activation until Bouchier-Hayes and colleagues cleverly adopted bimolecular fluorescence complementation (BiFC) to study caspase-2 dimerization (Bouchier-Hayes *et al*, 2009). In BiFC, complementary N-terminal and C-terminal halves of a fluorophore are fused to two

Department of Pharmacology and Cancer Biology, Duke University Medical Center, Durham, NC, USA
*Corresponding author. Tel: +1 919 684 2631; E-mail: sally.kornbluth@duke.edu
**Corresponding author. Tel: +1 919 613 8625; E-mail: kenkyo.matsuura@duke.edu
†These authors contributed equally to this work

interacting proteins (either different proteins or, as in the case of caspase-2, separate monomers of the same protein). When the proteins of interest interact they bring the nonfluorescent halves into close proximity and allow them to fold into a functional fluorophore. Thus, BiFC allowed for the identification of caspase-2-activating stimuli and provided insight into the timing and localization of caspase-2 dimers. We further speculated that this tool could be coopted for the identification of novel caspase-2 regulators.

In this study, we utilized the BiFC system to isolate the active caspase-2 complex. Specifically, we purified caspase-2 BiFC dimers by targeting the properly folded BiFC fluorophore with GFP-binding nanobodies (GFP-Trap). We applied this technique in the context of an inducible expression system to study caspase-2 dimerization at near endogenous levels. Interestingly, we found that caspase-2 dimers interact with TRAF1, TRAF2, and TRAF3, with TRAF2 being of particular importance for apoptotic stress-induced caspase-2 dimerization and subsequent apoptosis. Interestingly, TRAF2 had been previously reported to interact with caspase-2, though the nature of the interaction and functional role in caspase-2 regulation had not been delineated (Lamkanfi et al, 2005). We determined that TRAF2 binds directly with caspase-2 and that this is necessary for TRAF2 to promote caspase-2 activation. Furthermore, TRAF2, which has been shown to possess E3 ligase activity, was able to ubiquitylate caspase-2 and promote translocation of the active caspase-2 complex to a detergent-insoluble fraction. Finally, we identified three lysine residues in the caspase-2 prodomain (K15, K152, and K153) that are ubiquitylated after dimerization, modifications which promote further TRAF2 binding, caspase-2 dimer stability, and full caspase-2 activity.

## Results

### GFP-Trap selectively immunoprecipitates caspase-2 BiFC dimers to the exclusion of monomers

Given the paucity of data on mechanisms underlying caspase-2 activation, we sought to identify proteins interacting specifically with caspase-2 upon activation. We hypothesized that we could exploit the BiFC method to specifically isolate the active, dimerized form of caspase-2 along with any associated regulators through targeted pulldown of the caspase-2-linked Venus fluorophore reconstituted upon caspase-2 dimerization.

We therefore sought a GFP-targeting tool that would selectively capture the dimerized, reconstituted BiFC Venus to the exclusion of the monomeric BiFC Venus halves. For this, we employed GFP-Trap, a resin that binds GFP and GFP derivatives, including Venus. GFP-Trap utilizes a small (~15 kDa) antigen-binding domain, termed a nanobody, which is derived from an alpaca monoclonal antibody (Rothbauer et al, 2008). The antibody was designed against full-length GFP, and structural analysis of the GFP-Trap:GFP complex indicates that the nanobody recognition site spans both the VN and VC regions (corresponding to Venus residues 1–172 and 155–238, respectively) of the GFP molecule (Kubala et al, 2010). This led us to believe that it might selectively isolate dimerized BiFC Venus.

In order to test this, we transiently overexpressed, either individually or together, BiFC fragments composed of the caspase-2 prodomain (Casp2pro, residues 1–169; contains the CARD and known to be sufficient for activation-induced dimerization) fused with VN or VC. High caspase-2 protein levels are known to promote its spontaneous dimerization and activation in a manner dependent on its CARD-containing prodomain (Kumar et al, 1994; Wang et al, 1994; Butt et al, 1998; Colussi et al, 1998b). Thus, when the complementary Casp2pro BiFC fragments were co-overexpressed, Casp2pro BiFC signal was detected by flow cytometry, while the overexpression of Casp2pro-VN or Casp2pro-VC alone did not induce fluorescence (Fig 1A). The cells were then lysed and incubated with GFP-Trap, and the resultant precipitates analyzed by immunoblot. Excitingly, we found that GFP-Trap immunoprecipitated both fragments when they were co-overexpressed, but was unable to pull down either fragment alone (Fig 1B). This indicated that GFP-Trap only bound the reconstituted BiFC Venus, recognizing an epitope that is absent in the constituent fragments, and would thus be suitable for specifically isolating caspase-2 BiFC dimers.

Because of the link between high caspase-2 protein levels and spontaneous activation, we decided to construct a cell line with inducible expression of the caspase-2 BiFC system that would allow expression at near endogenous levels. The Casp2pro-VN and Casp2pro-VC coding sequences were cloned into a bidirectional tetracycline-controllable plasmid (pTRE-Tight-BI), which in turn was stably transfected into HeLa Tet-Off cells (Fig 1C). We then tested the utility of this cell line (hereafter referred to as Casp2pro BiFC cells) for detecting caspase-2 dimerization. In unstressed conditions, caspase-2 dimerization was not readily detected (Fig 1D). However, treatment with the DNA damaging agent cisplatin induced a significant BiFC Venus signal, which in turn could be blocked by shutting off the expression of the BiFC fragments with doxycycline.

Using this cell line, we then examined if GFP-Trap could immunoprecipitate low levels of caspase-2 BiFC dimers induced by known caspase-2 apoptotic stimuli (Bouchier-Hayes et al, 2009). Casp2pro BiFC cells were released from doxycycline and then treated with cisplatin, etoposide, or paclitaxel. Lysates of treated cells were probed with GFP-Trap and analyzed by immunoblot. Importantly, protein levels of the Casp2pro BiFC fragments were actually below levels of endogenous caspase-2 (thereby avoiding spontaneous dimerization), and treatment-induced BiFC dimers were readily precipitated by GFP-Trap (Fig 1E). GFP-Trap immunoprecipitated more Casp2pro BiFC fragments from cisplatin-treated cell lysates and less from untreated cell lysates compared with conventional antibody-based immunoprecipitation, further underscoring the specific capture ability of GFP-Trap for reconstituted BiFC Venus (Fig EV1A). Casp2pro BiFC protein levels were moderately increased when BiFC fluorescence was observed, but the mRNA levels remained unchanged, or even decreased as in the case of cisplatin treatment (Fig EV1B). We also observed that co-overexpression of the Casp2pro BiFC fragments yielded a similar increase in protein levels compared to individual overexpression of the constructs, indicating that the fragments are likely stabilized by dimerization (Fig 1B, input lanes).

Recently, it was reported that a failure of cytokinesis activates caspase-2 within the PIDDosome and induces p53-dependent cell cycle arrest (Fava et al, 2017). We therefore examined whether cytokinesis failure could induce Casp2pro dimerization. In p53

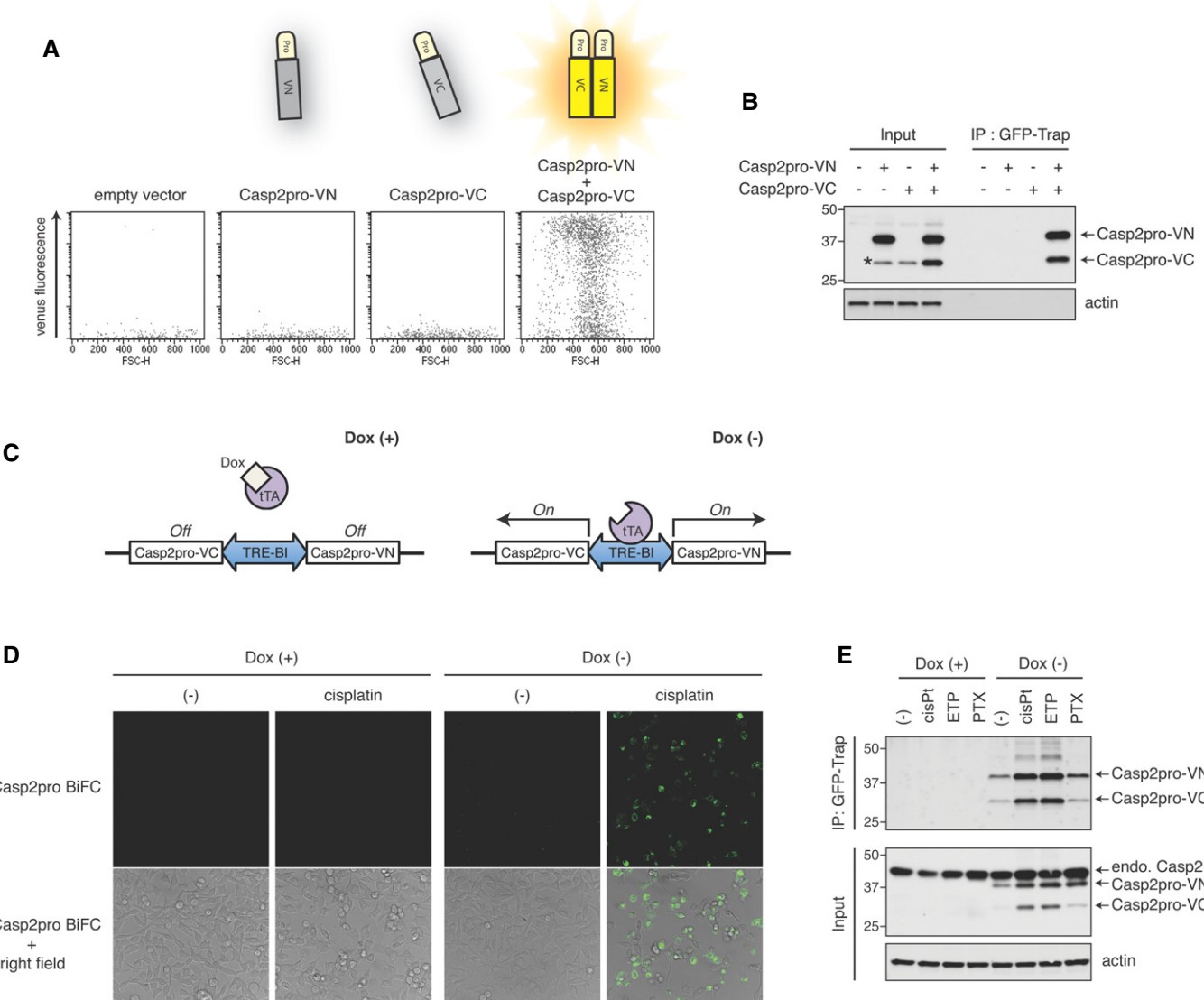

**Figure 1. GFP-Trap selectively immunoprecipitates caspase-2 BiFC dimers to the exclusion of monomers.**

A HEK293T cells were transfected with empty plasmid, Casp2pro-VN173 or/and Casp2pro-VC155 and cultured for 24 h, followed by flow cytometry to detect BiFC Venus signal.

B Lysates from (A) were immunoprecipitated (IP) using GFP-Trap, and Casp2pro BiFC dimers were detected with anti-caspase-2 (G310-1248) immunoblot (IB). The lower band (marked with asterisk) seen in lane 2 (Casp2pro-VN only) running at the same size as Casp2pro-VC is most likely partially degraded Casp2pro-VN. Molecular weight markers are indicated in kDa.

C Diagram of the bidirectional, tetracycline-controllable Casp2pro BiFC system: Doxycycline (Dox) binds and inhibits the tetracycline-controlled transactivator (tTA) and prevents it from inducing expression of the Casp2pro BiFC system. In the absence of Dox, expression of Casp2pro-VN173 and Casp2pro-VC155 is induced from the same promoter (TRE-BI).

D Representative confocal images of Casp2pro BiFC cells treated with or without 1 μg/ml Dox for 24 h, followed by 24 h with or without 20 μM cisplatin. All cells were also treated with pan-caspase inhibitor Q-VD(OMe)-OPh (10 μM). Scale bar indicates 50 μm.

E Casp2pro BiFC cells were treated with mock, 20 μM cisplatin (cisPt), 50 μM etoposide (ETP), or 100 nM paclitaxel (PTX) for 24 h in the presence of 10 μM Q-VD (OMe)-OPh, followed by GFP-Trap IP and anti-caspase-2 (G310-1248) IB. endo.; endogenous.

Source data are available online for this figure.

wild-type U-2OS and A549 cells, the Aurora kinase inhibitor ZM447439, which causes cytokinesis failure, was as effective as cisplatin at inducing a Casp2pro BiFC signal. In HeLa cells, ZM447439 also led to Casp2pro BiFC induction, albeit at lower levels than cisplatin (Fig EV1C). Another inducer of cytokinesis failure, DHCB (dihydrocytochalasin B), was also less effective than cisplatin at inducing Casp2pro BiFC in HeLa cells (Fig EV1D). Since HeLa cells lack a functional p53 response, it is possible that p53 status affects caspase-2 dimerization following cytokinetic failure.

## GFP-Trap immunoprecipitation of caspase-2 BiFC dimers identifies active caspase-2-interacting proteins

The ability to specifically capture caspase-2 dimers afforded an optimal system to isolate and characterize the interactome of dimerized caspase-2. Casp2pro BiFC cells were again treated with caspase-2-activating stressors, lysed, and incubated with GFP-Trap. To control for BiFC Venus-binding artifacts, we overexpressed unfused VN and VC, which can drive spontaneous dimerization, in Casp2pro BiFC cells treated with doxycycline. These cells were treated similarly to the experimental samples, lysed, and then combined before GFP-Trap immunoprecipitation (Fig EV2A). The resulting precipitates were confirmed by SDS–PAGE and silver staining (Fig 2A) and then analyzed by mass spectrometry for protein identification.

This strategy identified several apoptosis-related proteins as putative caspase-2 dimer binders: RAIDD, TRAF1, TRAF2, TRAF3, HtrA2, and cIAP1 (Fig 2B). The identification of RAIDD, a well-known caspase-2 binding partner, confirmed the validity of the approach. Interestingly, PIDD was not identified by our proteomics screen of caspase-2 interactors, and knockdown of PIDD had no effect on cisplatin-induced caspase-2 dimerization in Casp2pro BiFC cells (Fig EV2B and C). Furthermore, RAIDD knockdown affected neither caspase-2 dimerization nor apoptosis in response to cisplatin (Fig EV2D–F). These results suggest that the PIDDosome is not involved in caspase-2 dimerization in this context. Due to the prevalence of TRAF family members binding to caspase-2, and because TRAF proteins are known to act as adaptors for signaling complexes, we decided to focus our efforts on TRAF1, 2, and 3.

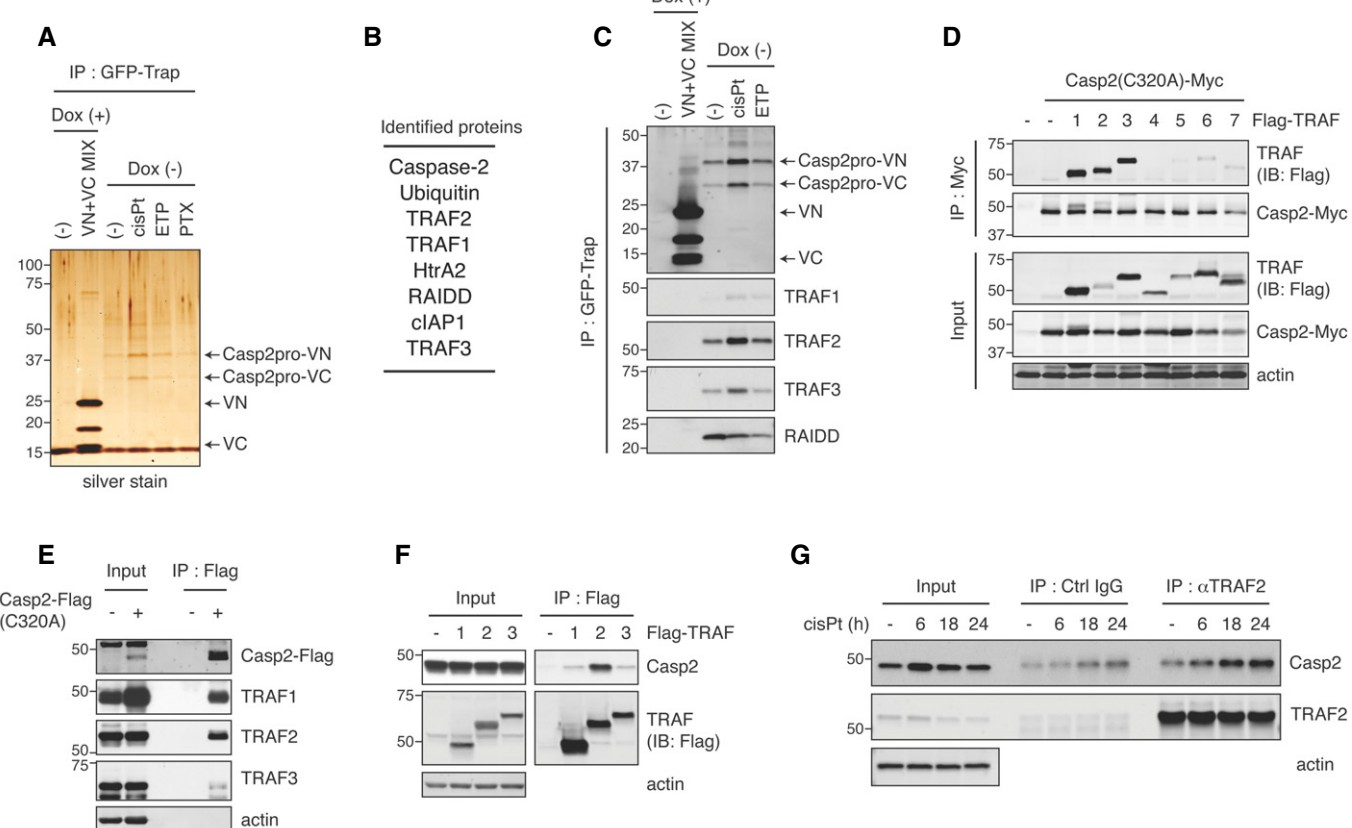

**Figure 2.   GFP-Trap immunoprecipitation of caspase-2 BiFC dimers identifies active caspase-2-interacting proteins.**

A  Casp2pro BiFC cells were treated with mock, 20 μM cisplatin, 50 μM etoposide, or 100 nM paclitaxel for 24 h in the presence of 10 μM Q-VD(OMe)-OPh, followed by GFP-Trap IP and silver stain to examine immunoprecipitates. For a negative control, Casp2pro BiFC expression was shut off by Dox and substituted with unconjugated VN-VC BiFC dimers pooled from cells treated similarly to experimental samples. The procedure outline is in Fig EV2A.

B  Casp2pro BiFC dimer-interacting proteins identified by mass spectrometric analysis were listed in descending order of peptide abundance.

C  Casp2pro BiFC cells were treated with or without 1 μg/ml Dox for 24 h, then mock, 20 μM cisplatin, or 50 μM etoposide for 24 h in the presence of 10 μM Q-VD(OMe)-OPh, followed by GFP-Trap IP and IB. BiFC fragments were detected with anti-GFP (FL, rabbit polyclonal antibody) IB.

D  Casp2(C320A)-Myc and Flag-tagged TRAF1-7 were co-expressed in HEK293T cells for 48 h, followed by anti-Myc IP and IB.

E  Casp2(C320A)-Flag was transfected into HeLa cells and expressed for 48 h, followed by anti-Flag IP and IB.

F  Flag-TRAF1-3 were transiently expressed in HeLa cells for 48 h followed by anti-Flag IP and IB.

G  HeLa cells were treated with 20 μM cisplatin in the presence of 10 μM Q-VD(OMe)-OPh for indicated periods. Lysates were prepared and immunoprecipitated with anti-TRAF2 antibody or control IgG, followed by IB.

Source data are available online for this figure.

Immunoblot analysis of GFP-Trap pulldowns confirmed that endogenous TRAF1, 2, and 3 bound to Casp2pro BiFC dimers and that this binding was increased by caspase-2-activating stimuli, especially cisplatin (Fig 2C). There are seven members of the TRAF family (Xie, 2013); in order to determine the exclusivity of TRAF1, 2, and 3 in caspase-2 regulation, we co-transfected a full-length Casp2-Myc, in which the catalytic cysteine was mutated to alanine (C320A) to prevent induction of cell death, with each of the seven TRAFs and then immunoprecipitated by anti-Myc agarose beads. Immunoblot analysis revealed that only TRAF1, 2, and 3 bound significantly to caspase-2 (Fig 2D). We also found that overexpressed caspase-2, which spontaneously dimerizes, bound endogenous TRAF1, 2, and 3 (Fig 2E). We performed the reverse experiment, pulling down on overexpressed TRAF1, 2, or 3, and found that only TRAF2 bound strongly to endogenous caspase-2, possibly indicating a higher specificity of interaction with caspase-2 than the other TRAFs (Fig 2F). Among CARD-containing caspases, overexpressed TRAF2 could interact with caspase-9 (and caspase-2), but not with caspase-1, 4, or 5 (Fig EV2G). Finally, we confirmed the interaction of endogenous TRAF2 and caspase-2, which was increased after cisplatin treatment in a time-dependent manner (Fig 2G).

The previously reported complex containing TRAF2 and caspase-2 was found to be an activator of the NF-κB pathway (Lamkanfi *et al*, 2005). Consistent with those findings, caspase-2(C320A) overexpression induced phosphorylation of IκBα and p65 and conversion of NF-κB1 p105 to p50, indicative of NF-κB pathway activation (Fig EV2H). Cisplatin treatment slightly activated the NF-κB pathway (Fig EV2I-left), but it was a considerably weaker inducer than caspase-2(C320A) overexpression or treatment with TNFα, a well-known NF-κB pathway activator (Fig EV2H and I-right). However, inhibitors of the NF-κB pathway had no effect on cisplatin-induced Casp2pro BiFC dimer formation, and therefore, it is unlikely that any effects of TRAF2 on caspase-2 activation are exerted indirectly via NF-κB signaling in this context (Fig EV2J).

## TRAF2 is required for caspase-2-initiated apoptosis

In order to understand the functional relevance of these caspase-2/TRAF interactions, we knocked down the TRAFs in the Casp2pro BiFC cells and looked at the effect on caspase-2 dimerization. TRAF2 knockdown significantly decreased caspase-2 dimerization in response to cisplatin (Fig 3A and B). However, TRAF3 knockdown only partially reduced caspase-2 dimerization, while loss of TRAF1 slightly increased it (Fig 3B). TRAF2 knockdown in Casp2pro BiFC cells also decreased caspase-2 dimerization in response to etoposide and paclitaxel treatment, suggesting a general regulation of caspase-2 dimerization by TRAF2 (Fig EV3A). We then examined how the absence of the TRAFs affected apoptosis induction. Stable knockdown of TRAF2 using two separate shRNA significantly blocked cisplatin-induced apoptosis (Fig 3C). Consistent with a role for TRAF2 in caspase-2 regulation, stable knockdown of caspase-2 yielded similar results to TRAF2 knockdown (Fig 3D). The importance of caspase-2 and TRAF2 for apoptosis induced by cisplatin, etoposide, and paclitaxel was also confirmed in the breast cancer cell line BT474 (Fig EV3B). On the other hand, knockdown of TRAF3 only partially blocked cell death (Fig 3E), while targeting TRAF1 had no effect (A. C. Robeson, unpublished observations). The effect of TRAF2 knockdown on cisplatin-induced apoptosis was confirmed by a reduction in the cleavage of caspase-2, caspase-9 (Fig 3F), and the executioner caspase-3 (Fig 3G). TRAF2 knockdown also reduced the oligomerization of endogenous caspase-2, as confirmed by crosslinking with BMH reagent (Fig EV3C). Critically, TRAF2 knockdown also prevented cleavage of Bid, a known target of caspase-2, as well as downstream activation of Bax (Fig 3H and I). Together, these data indicate that TRAF2 controls the initiation of apoptosis by promoting mitochondrial outer membrane permeabilization via caspase-2 cleavage of tBid, while TRAF3 and TRAF1 play a lesser role.

To further confirm TRAF2 involvement in caspase-2 activation, we generated TRAF2 knockout cells by CRISPR technology. Surprisingly, TRAF2 knockout did not affect caspase-2 dimerization in HeLa cells (Fig EV3D). TRAF2 knockout also had no effect on cell death (Fig EV3E). Interestingly, we found that TRAF3 was upregulated in TRAF2 knockout cells, but not in shRNA-mediated TRAF2 knockdown cells (Fig EV3F). TRAF2 knockout clonal cells isolated from single colonies showed even more obvious TRAF3 upregulation (Fig EV3G). This is reminiscent of a previous report that gene disruptions, but not gene knockdowns, can lead to compensatory mechanisms, such as upregulation of other genes in the same signaling pathway (Rossi *et al*, 2015). Since TRAF3 is also involved in caspase-2 dimerization and activation (Fig 3B and E), it is quite possible that TRAF3 is compensating the loss of TRAF2. Supporting this hypothesis, TRAF3 knockdown reduced caspase-2 dimerization in TRAF2 knockout cells, but had a marginal effect on control cells (Fig EV3H).

**Figure 3.  TRAF2 is required for caspase-2-initiated apoptosis.**

A    Representative confocal images. Casp2pro BiFC cells were transfected with control siRNA (siCtrl) or siRNA targeting TRAF2 (siTRAF2) for 48 h and then treated with 20 μM cisplatin for 24 h in the presence of 10 μM Q-VD(OMe)-OPh. Scale bar indicates 50 μm.

B    Casp2pro BiFC cells were transfected with the indicated siRNA for 48 h and then treated with 20 μM cisplatin in the presence of 10 μM Q-VD(OMe)-OPh for 24 h. BiFC signal was detected by flow cytometry. *n* = 3 independent experiments (means + s.e.m.). **P* < 0.05 by unpaired two-tailed *t*-test.

C    shNT (non-targeting) or shTRAF2 HeLa cells were treated with 20 μM cisplatin for 24 h, followed by annexin V staining and flow cytometry analysis. *n* = 3 independent experiments (means + s.e.m.). **P* < 0.005 by unpaired two-tailed *t*-test.

D    shNT or shCasp2 HeLa cells were analyzed as in (C). *n* = 3 independent experiments (means + s.e.m.). **P* < 0.001 by unpaired two-tailed *t*-test.

E    HeLa cells were transfected with siRNA targeting TRAF3 for 48 h, followed by 20 μM cisplatin treatment for 24 h, then stained with annexin V and analyzed by flow cytometry. *n* = 4 independent experiments (means + s.e.m.). **P* < 0.05 by unpaired two-tailed *t*-test. (A–E, right panel) Knockdown efficiency was assessed by IB.

F–I    shNT or shTRAF2 #1 HeLa cells were treated with 20 μM cisplatin for 24 h (F and G) or 18 h (H and I), followed by IB for caspase-2 or caspase-9 (F), caspase-3 (G), or BID (H). Active Bax was immunoprecipitated by active Bax-specific antibody (6A7) and analyzed by IB with total anti-Bax antibody (I).

Source data are available online for this figure.

## TRAF2 interacts directly with caspase-2 to induce activation

Several TRAF-interacting motif consensus sequences have been previously identified, to include (P/S/A/T-X-Q/E-E) and (P/H-V/I/ T-Q-E-T/S) (Pullen *et al*, 1999; Ye *et al*, 1999). A scan of the caspase-2 sequence revealed a putative TRAF-interacting motif (TIM) at residues 247–251 – TAQEM (Fig 4A). To investigate the importance of this sequence, we constructed a caspase-2 mutant

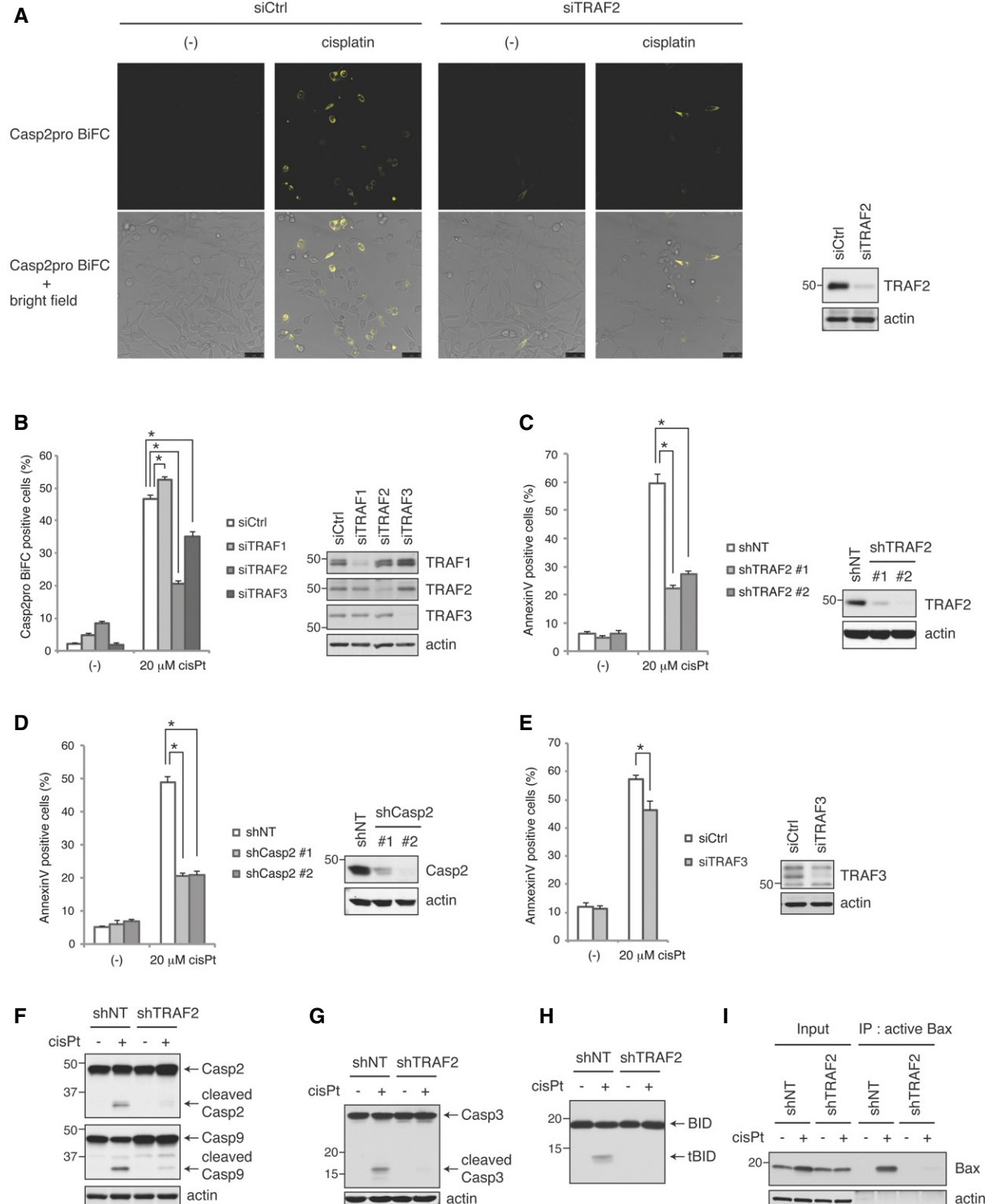

**Figure 3.**

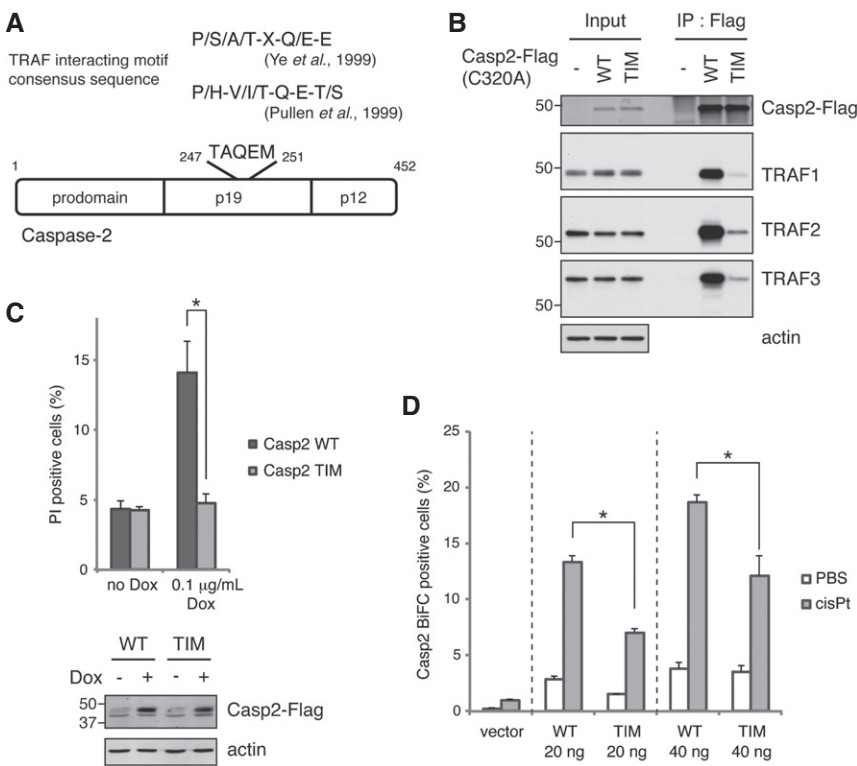

**Figure 4.  TRAF2 interacts directly with caspase-2 to induce activation.**

A   A diagram of the caspase-2 protein showing the major domains and the location of a putative TRAF-interacting motif (TIM).

B   Casp2(C320A)-Flag plasmids (wild type or TIM mutant) were expressed in HeLa cells for 24 h followed by anti-Flag IP and IB.

C   Stably transfected, catalytically active caspase-2 (wild type or TIM mutant) was induced in HeLa Tet-On cells with 0.1 µg/ml doxycycline for 24 h, followed by propidium iodide (PI) staining and flow cytometry. *n* = 3 independent experiments (means + s.e.m.). *P < 0.05 by unpaired two-tailed *t*-test (top). Casp2-Flag expression was confirmed by IB (bottom).

D   Indicated amounts of Casp2(C320A) BiFC constructs were transiently transfected, and cells were cultured for 24 h followed by 20 µM cisplatin treatment in the presence of 10 µM Q-VD(OMe)-OPh for 24 h, and then, BiFC signal was detected by flow cytometry. *n* = 3 independent experiments (means + s.e.m.). *P < 0.05 by unpaired two-tailed *t*-test.

Source data are available online for this figure.

where all of the TIM residues were changed to alanine [hereafter referred to as Casp2(TIM)]. A pulldown experiment revealed that mutation of the TIM reduced caspase-2 binding to all three TRAFs (Fig 4B). To examine the effect of the loss of the TIM on caspase-2 apoptosis-inducing activity, HeLa Tet-On cells were transduced with retroviral particles containing tetracycline-inducible constructs of Casp2(wild type) and (TIM). Expression of Casp2(wild type) readily induced cell death but Casp2(TIM) expression had no effect (Fig 4C). Furthermore, we constructed Casp2 BiFC fragments with mutant TIMs and examined the effect on dimerization. HeLa cells were carefully transfected with low amounts of the plasmids to reduce spontaneous dimerization and then treated with cisplatin. We found that mutation of the TIM significantly reduced cisplatin-induced caspase-2 dimerization (Fig 4D). These data strongly suggest that loss of TRAF interactions impedes the death-inducing function of caspase-2.

It is possible that the TIM mutation might disrupt the caspase-2 structure directly, impeding its biological activity. We therefore examined the interaction of the TIM mutant with RAIDD, a known caspase-2 binding protein. We found that the caspase-2 TIM mutant could bind RAIDD as efficiently as wild-type caspase-2 (Fig EV4A), suggesting that the TIM mutant did not disrupt overall protein structure. Since TRAF2

was able to bind to Casp2pro BiFC (Fig 2A–C), it seemed likely that TRAF2 had the ability to bind outside the TIM, even if the TIM was critical for biological activity. That said, it remained possible that the transfected prodomain was interacting with endogenous caspase-2 to confer TRAF2 binding through the endogenous TIM. To exclude the involvement of endogenous caspase-2, a caspase-2 3′-UTR-targeting siRNA was used, and mVenus-tagged caspase-2—full-length, prodomain, or Δ1–169 (no prodomain, with an intact TIM)—was expressed. As shown in Fig EV4B, the prodomain still bound TRAF2 without endogenous caspase-2, but binding was nearly abolished in the Δ1–169 mutant. These data suggest that both the prodomain and TIM contribute to TRAF binding. And, since mutation of either domain in the context of the full-length caspase-2 impedes binding, both regions are necessary for optimal binding of TRAF2 and optimal biological activity.

## Dimerized caspase-2 is ubiquitylated in a TRAF2-dependent manner at K15, K152, and K153, which in turn promotes further TRAF2 binding in a positive feedback loop

There are several potential mechanisms by which TRAF2 might control caspase-2 activation: TRAF2 is known to act as a scaffold

that mediates the interaction with other proteins, but it also possesses an amino-terminal RING domain that has been shown to confer E3 ligase activity (Alvarez *et al*, 2010; Gonzalvez *et al*, 2012). Indeed, ubiquitin was identified by mass spectrometry as one of the major proteins pulled down with the Casp2pro BiFC dimers (Fig 2B). Interestingly, when we performed GFP-Trap immunoprecipitation after cisplatin treatment, we found that the Casp2pro BiFC

fragments were ubiquitylated (Fig 5A). We also performed a denaturing/renaturing immunoprecipitation to examine ubiquitylation on endogenous caspase-2. Specifically, the samples were boiled in the presence of SDS to disrupt non-covalent interactions, then diluted in lysis buffer and subjected to immunoprecipitation. Consistent with our earlier results, we found that caspase-2 ubiquitylation was increased in response to cisplatin treatment (Fig 5B). We also

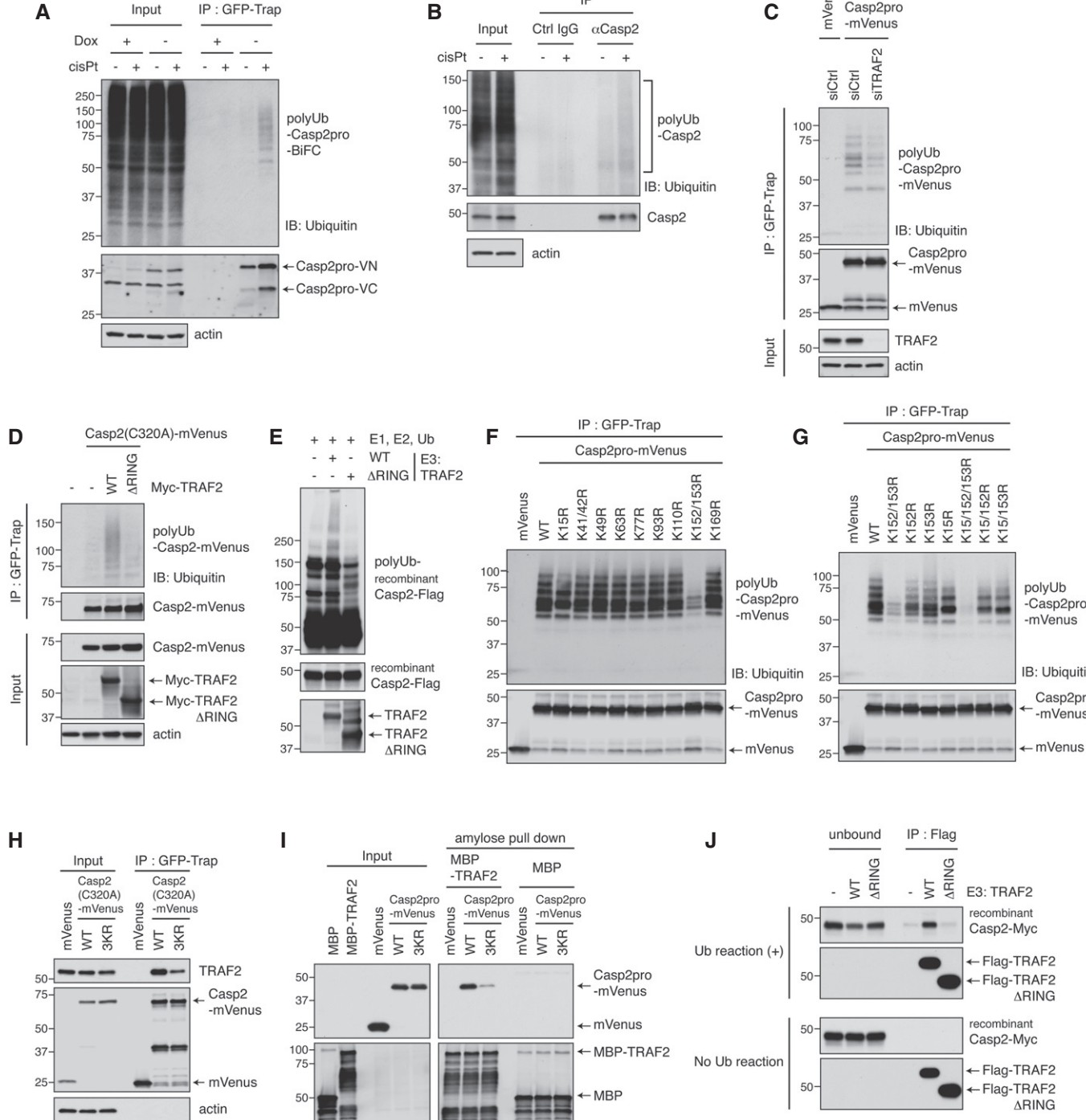

**Figure 5.**

                                             

observed ubiquitylation of the overexpressed caspase-2 prodomain in several cell lines (Fig EV5A). Interestingly, the ubiquitylation of caspase-2 prodomain was diminished by TRAF2 knockdown (Fig 5C). TRAF2 might positively regulate caspase-2 ubiquitylation indirectly by protecting it from deubiquitylation. To test this, Casp2pro-mVenus ubiquitylated with HA-ubiquitin was purified by GFP-Trap from HeLa cells and applied to an *in vitro* deubiquitylation assay. A rapid reduction of caspase-2 polyubiquitylation was observed, but the addition of recombinant TRAF2 failed to reverse this trend (Fig EV5B). In contrast, overexpression of a wild-type TRAF2 induced caspase-2 ubiquitylation, while a mutant of TRAF2 lacking the RING domain failed to do the same (Fig 5D). Importantly, TRAF2 was able to ubiquitylate recombinant caspase-2 *in vitro* in a manner dependent on its RING domain (Fig 5E).

This led us to hypothesize that TRAF2 acts on caspase-2 to promote its activation through non-degradative ubiquitylation of the prodomain. We thus created arginine mutants for each lysine residue in the prodomain in order to determine which of them contribute to caspase-2 ubiquitylation. Two mutants yielded a significant phenotype: The K152/153R mutation dramatically reduced ubiquitylation of the caspase-2 prodomain, while the K15R mutation had a more moderate effect (Fig 5F). When all three lysines were mutated in the same protein (hereafter referred to as 3KR), caspase-2 prodomain ubiquitylation was almost completely abolished (Fig 5G). This was confirmed by a reciprocal IP method: HA-ubiquitin and Casp2pro-mVenus (wild type or 3KR) were co-transfected into cells, and lysates were immunoprecipitated by anti-HA affinity beads. The 3KR mutation almost completely abolished the ubiquitylation of the caspase-2 prodomain (Fig EV5C). Moreover, ubiquitin linkage-specific antibodies revealed that Casp2pro-mVenus underwent both K48- and K63-mediated polyubiquitylation (Fig EV5D).

Interestingly, we found that the 3KR mutation diminished the binding of caspase-2 to endogenous TRAF2 (Fig 5H). To confirm this, we overexpressed mVenus-tagged caspase-2 prodomain to promote its ubiquitylation and then mixed in bacterially expressed recombinant MBP-tagged TRAF2. Recombinant TRAF2 was then retrieved by amylose resin, and the binding of caspase-2 was assessed. Wild-type caspase-2 bound recombinant MBP-TRAF2, but the 3KR mutant of caspase-2 showed much weaker binding (Fig 5I). Finally, for a more direct validation that ubiquitylation of caspase-2

enhances its interaction with TRAF2, we performed an *in vitro* ubiquitylation assay of caspase-2 as before (Fig 5E), followed by an *in vitro* binding assay. Wild-type TRAF2 strongly bound recombinant caspase-2 after ubiquitylation, but the RING domain mutant was unable to do the same (Fig 5J). Together these findings indicate that the ubiquitylation of caspase-2 by TRAF2 promotes further TRAF2 binding in a positive feedback loop.

### TRAF2 shifts active, dimerized caspase-2 to a detergent-insoluble fraction in a RING domain-dependent manner

In seeking to identify a biochemical correlate of TRAF2's ability to promote caspase-2 ubiquitylation, we examined the localization of caspase-2 following overexpression of TRAF2 or its RING domain mutant. Previous studies found that caspase-2 localizes predominantly to the nucleus (Colussi *et al*, 1998a; Paroni *et al*, 2002; Baliga *et al*, 2003), but its dimerization occurs in the cytoplasm (Bouchier-Hayes *et al*, 2009). In line with this, mVenus-tagged caspase-2 localized mostly in the nucleus, but then formed punctate structures in the cytoplasm when TRAF2 was overexpressed (Fig 6A). In contrast, the TRAF2 RING domain mutant failed to induce foci formation of caspase-2, although caspase-2 was still localized to the cytoplasm (Fig 6A). Cisplatin-induced caspase-2 BiFC dimerization was also primarily detected in the cytoplasm (Figs 1D and 3A), with some cells showing a punctate signal similar to that induced by TRAF2 overexpression, as shown in Fig 6A. The presence of large punctate structures suggested that TRAF2 overexpression might alter caspase-2 subcellular compartmentalization. Interestingly, biochemical fractionation revealed that caspase-2 was sequestered into a 0.1% NP-40-insoluble fraction by TRAF2 overexpression in a RING domain-dependent manner (Fig 6B). Overexpression-induced Casp2pro BiFC dimerization was also found mostly in the insoluble fraction, while dimerization of the BiFC fragments lacking the caspase-2 sequence (VN and VC) occurred in the soluble fraction (Fig EV6A). When the same fractionation approach was applied to Casp2pro BiFC cells, we found that cisplatin-induced caspase-2 dimers were primarily localized in the insoluble fraction and were heavily ubiquitylated (Fig 6C). In addition, both overexpressed full-length caspase-2(C320A)-mVenus and endogenous caspase-2 were more heavily ubiquitylated in the insoluble fraction in response to cisplatin treatment (Fig EV6B and C). These results suggest that



**Figure 5.   Dimerized caspase-2 is ubiquitylated in a TRAF2-dependent manner at K15, K152, and K153, which in turn promotes further TRAF2 binding in a positive feedback loop.**

A    Casp2pro BiFC cells were treated with 20 μM cisplatin for 24 h in the presence of 10 μM Q-VD(OMe)-OPh, followed by GFP-Trap IP and IB with anti-ubiquitin or anti-GFP antibody.

B    HeLa cells were treated with 20 μM cisplatin for 24 h in the presence of 10 μM Q-VD(OMe)-OPh. Lysates were denatured/renatured and immunoprecipitated with anti-caspase-2 antibody or control IgG, followed by IB with anti-ubiquitin or anti-caspase-2 antibody.

C    HeLa cells were transfected with TRAF2 siRNA for 24 h, then transfected with Casp2pro-mVenus for 48 h, followed by GFP-Trap IP and IB.

D    Casp2(C320A)-mVenus was co-expressed with the indicated TRAF2 constructs and then pulled down with GFP-Trap and analyzed by IB.

E    *In vitro* ubiquitylation of recombinant Casp2-Flag by Myc-TRAF2 (wild type or ΔRING) purified from HEK293T cells.

F, G   Casp2pro-mVenus wild type and indicated lysine mutants were expressed for 24 h in HEK293T cells, followed by GFP-Trap IP and IB.

H    HEK293T cells were transfected with Casp2(C320A)-mVenus (wild type or K15/152/153R (3KR) mutant) constructs for 48 h, followed by GFP-Trap IP and IB.

I    HeLa cells were transfected with Casp2pro-mVenus for 48 h and lysed. Recombinant MBP-TRAF2 or MBP control proteins were incubated in the lysate for 1 h, followed by amylose pulldown and IB to detect caspase-2 binding.

J    *In vitro* ubiquitylation was performed as in (E), with recombinant Casp2-Myc protein and Flag-TRAF2 (wild type or ΔRING) purified from HEK293T cells. After 3-h incubation at 37°C (Ub reaction (+)) or on ice (No Ub reaction), the reaction was incubated with anti-Flag beads. Immunoprecipitated and unbound fractions were analyzed by IB.

Source data are available online for this figure.

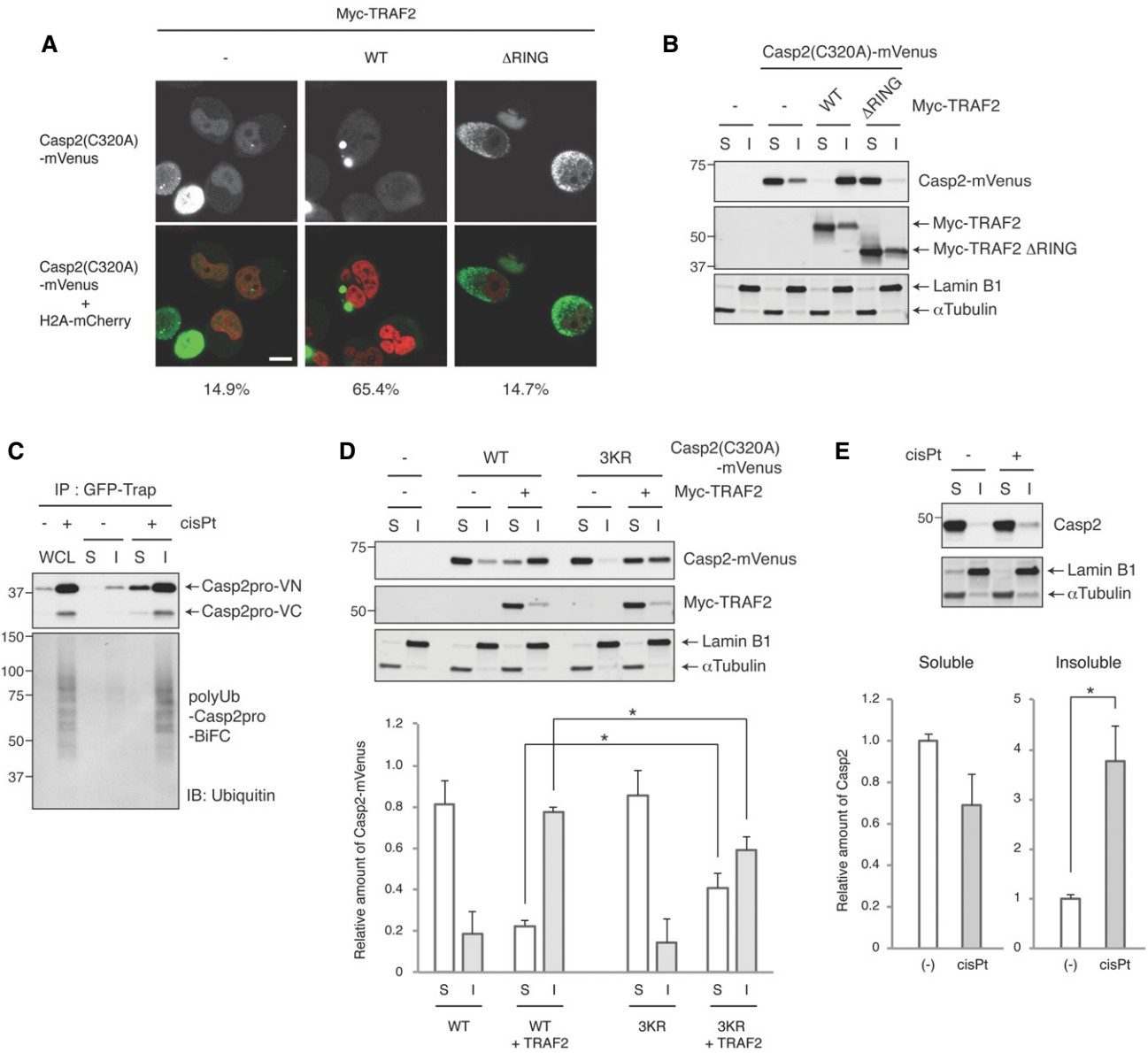

**Figure 6. TRAF2 shifts active, dimerized caspase-2 to a detergent-insoluble fraction in a RING domain-dependent manner.**

A   Representative confocal images. HeLa cells were co-transfected with Casp2(C320A)-mVenus and Myc-TRAF2 (wild type or RING domain mutant (ΔRING)) and then allowed to express for 48 h before imaging to detect caspase-2 localization. Nuclei were visualized by co-transfection of H2A-mCherry. Scale bar indicates 10 μm. The percentage of cells with caspase-2 punctate signals was quantified and listed below each image. At least 40 cells were counted from $n = 5$ independent experiments.

B   HeLa cells were co-transfected with Casp2(C320A)-mVenus and the indicated Myc-TRAF2 constructs for 48 h and then biochemically fractionated (S: soluble, I: insoluble fraction) to monitor caspase-2 localization in 0.1% NP-40-soluble or -insoluble fractions.

C   Casp2pro BiFC cells were treated with 20 μM cisplatin in the presence of 10 μM Q-VD(OMe)-OPh for 24 h and then biochemically fractionated as in (B) or lysed as whole cell lysate (WCL). Lysates were subjected to GFP-Trap IP and analyzed by IB.

D   HeLa cells were co-transfected with Casp2(C320A)-mVenus (wild type or 3KR mutant) and Myc-TRAF2, and cultured for 48 h. Then, cells were biochemically fractionated as in (B) and analyzed by IB to monitor Casp2(320A)-mVenus localization (top). Casp2(320A)-mVenus protein level was quantified and plotted. $n = 4$ independent experiments (means + s.e.m.). *$P < 0.05$ by unpaired two-tailed $t$-test (bottom).

E   HeLa cells were treated with or without 20 μM cisplatin in the presence of 10 μM Q-VD(OMe)-OPh for 24 h and then biochemically fractionated as in (B) to monitor endogenous caspase-2 localization (top). Caspase-2 protein level was quantified and plotted. $n = 4$ independent experiments (means + s.e.m.). *$P < 0.01$ by unpaired two-tailed $t$-test (bottom).

Source data are available online for this figure.

both caspase-2 ubiquitylation and localization correlate with activation. Along these lines, we found that TRAF2 overexpression-induced caspase-2 translocation into the insoluble fraction was partially suppressed by the 3KR mutation (Fig 6D). Furthermore, endogenous caspase-2 was localized in the soluble fraction under unstressed conditions, but translocated to the insoluble fraction in

response to cisplatin treatment (Fig 6E). Thus, TRAF2-mediated ubiquitylation seems to regulate the transfer of caspase-2 to a detergent-insoluble fraction in correlation with its activation.

## Caspase-2 prodomain ubiquitylation promotes dimer stability and activity to induce apoptosis

To further examine the function of TRAF2-mediated caspase-2 ubiquitylation, the 3KR mutations were introduced into the caspase-2 BiFC system. Surprisingly, we did not observe a significant difference in the percentage of BiFC-positive cells when comparing the wild-type and 3KR mutant (Fig 7A). However, we

found that when we attempted to immunoprecipitate the BiFC dimers, GFP-Trap pulled down significantly less 3KR BiFC dimers compared to wild type, suggesting that the non-ubiquitylatable caspase-2 might form an unstable dimer (Fig 7B). To further investigate this possibility, we co-expressed Myc-Casp2pro and Casp2pro-mVenus at low levels in HeLa cells, treated with cisplatin, immunoprecipitated the resulting heterodimers with GFP-Trap, and then subjected them to salt washes of increasing stringencies (Fig 7C). This assay revealed that Casp2(3KR) dimers are less stable and more easily disrupted, while wild-type caspase-2 dimers remain stable even under high concentrations of salt wash up to 750 mM NaCl (Fig 7C).

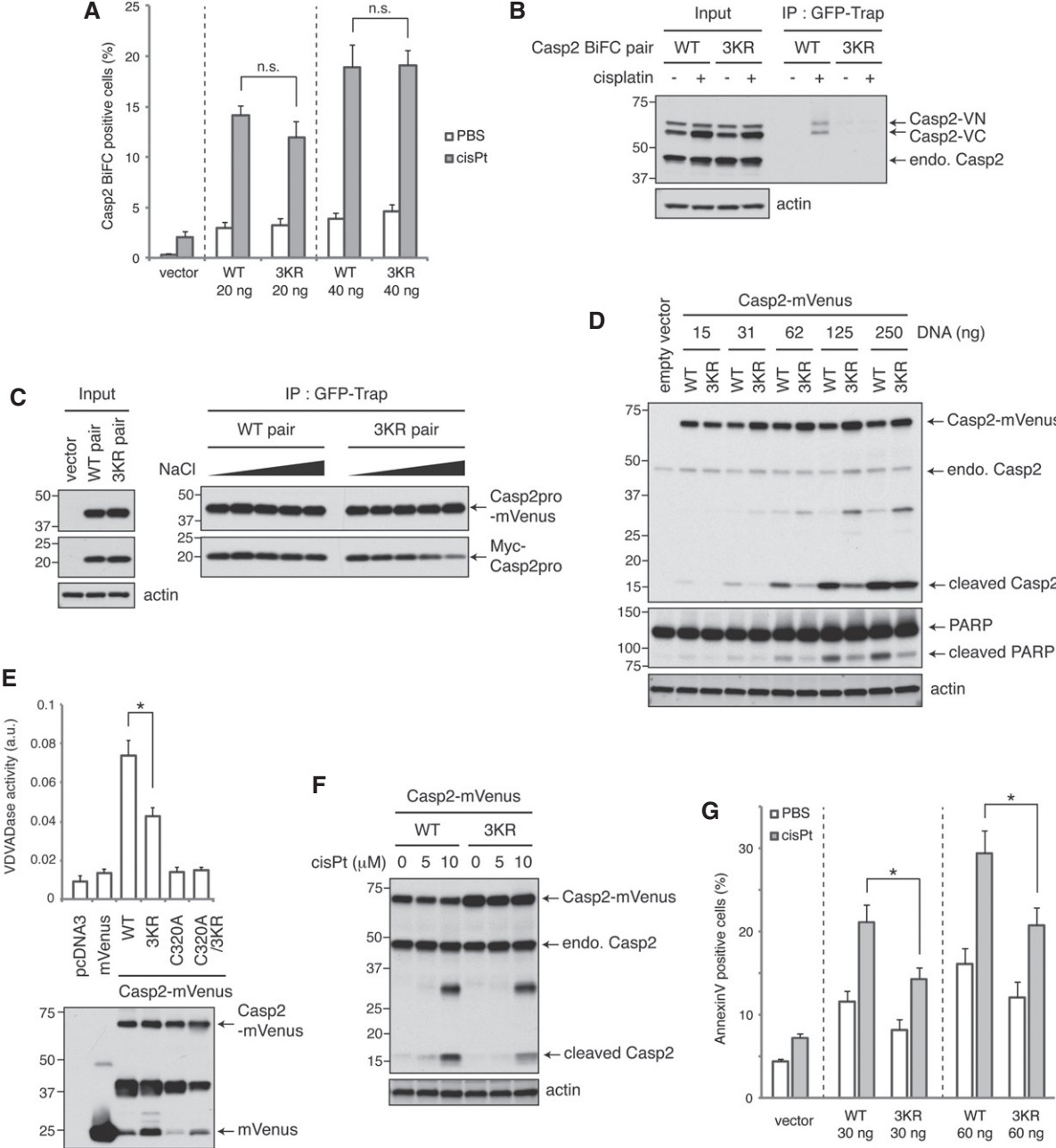

**Figure 7.**

We then examined how the loss of these three ubiquitylation sites affects caspase-2 function. When we expressed increasing amounts of catalytically active caspase-2, we found that wild-type caspase-2 exhibited higher activity than the 3KR mutant, as evidenced by autoprocessing (Fig 7D). This effect on catalytic activity was validated in an enzymatic assay, which showed that the 3KR mutations significantly reduced the ability of caspase-2 to cleave the artificial substrate VDVAD-pNA (Fig 7E). Moreover, cisplatin-induced cleavage of the 3KR mutant was less than the wild type when caspase-2 was expressed at a near endogenous level (Fig 7F). Finally, the 3KR mutation significantly retarded the induction of apoptosis by cisplatin (Fig 7G). We also performed this experiment in caspase-2 knockdown cells with an shCasp2-resistant caspase-2 addback and observed a similar level of reduction in caspase-2 cleavage and cell death induced by the 3KR mutation (Fig EV7A). Further supporting the role of TRAF2-mediated caspase-2 regulation in apoptosis, TRAF2 overexpression induced cell death in a RING domain-dependent manner (Fig EV7B), and this cell death was suppressed by caspase-2 knockdown (Fig EV7C). Together these data indicate that K15, K152, and K153, which are subject to ubiquitylation by TRAF2, play a critical role in stabilizing caspase-2 dimerization to promote its full pro-apoptotic activation.

## Discussion

The ability to study protein complexes is often hampered by a lack of precise tools. In this study, we demonstrated that the combination of BiFC with GFP-Trap nanobodies produces a powerful tool for isolating and characterizing caspase dimers and their interactomes. Although we only used this method to investigate caspase-2 complexes, the modular nature of the BiFC system should be broadly useful. Indeed, a similar technique, termed bimolecular complementation affinity purification (BiCAP), was recently reported by Croucher and colleagues in a study of ERBB2 dimers, showing that it can be applied to a wide variety of dimer-based complexes (Croucher *et al*, 2016). The advantage of this method is that it enables quick and simple purification of dimer complexes of interest, as opposed to the conventional dimer complex

immunoprecipitation, which requires sequential purification of two different tags. Also, the high specificity of GFP-Trap against GFP derivatives eliminates non-specific binding of unrelated proteins, which is often seen by conventional antibody-based immunoprecipitation. And since the proportion of dimer to monomer in cells could be low, as indicated for active caspase-2 by gel filtration experiments from previous studies (Manzl *et al*, 2009; Ando *et al*, 2012), the capacity of this method to capture endogenous levels of dimer makes it ideal for mass spectrometry studies.

Using this approach, we showed that caspase-2 dimers specifically interacted with TRAF1, TRAF2, and TRAF3 but not other members of the TRAF family (Fig 2). This is the first time that TRAF3 has been shown to interact with caspase-2, while TRAF1 and TRAF2 had been previously identified as caspase-2 interactors by Lamkanfi *et al* (2005). In that study, caspase-2 and TRAF2, in complex with RIPK1, were found to positively regulate NF-κB signaling, which is the canonical TRAF2 pathway. However, these complexes were characterized in the context of cell lysates heated to 37°C, rather than in the context of known stress-inducing caspase-2 stimuli. It is unclear if this *in vitro* technique models a physiologically relevant stimulus, though it may be consistent with a role for caspase-2 in the heat shock response. Furthermore, we were unable to identify RIPK1 as a binding partner in either our proteomics experiments or through immunoblot analysis (K. Matsuura, unpublished observations). It seems reasonable that different stimuli promote distinct caspase-2/TRAF2 complexes, with some perhaps functioning in a non-apoptotic capacity. Indeed, this might also be the case for TRAF1 or TRAF3, as they did not seem to be as critical for DNA damage-induced apoptosis (Fig 3E and A. C. Robeson, unpublished observations).

In this study, we demonstrated that TRAF2 is required for caspase-2 dimerization and consequent cell death (Fig 3). Pathway analysis confirmed that TRAF2 acts upstream of the mitochondria to trigger cell death (Fig 3F–I), confirming the role of the caspase-2/TRAF2 complex in initiating apoptosis. We observed that TRAF2 bound directly to caspase-2 at least in part through a TRAF-interacting motif (TIM) and that this sequence is important for caspase-2 dimerization and cell death (Fig 4). This was somewhat puzzling since the TIM lies outside of the caspase-2 prodomain, yet the

◄ **Figure 7. Caspase-2 prodomain ubiquitylation promotes dimer stability and activity to induce apoptosis.**

A    Various amounts of the indicated Casp2(C320A) BiFC constructs were transfected into HeLa cells and allowed to express for 24 h. Cells were then treated with 20 μM cisplatin in the presence of 10 μM Q-VD(OMe)-OPh for 24 h, and BiFC was detected by flow cytometry. *n* = 3 independent experiments (means + s.e.m.). *P* > 0.05 (0.30 (20 ng) or 0.94 (40 ng)) by unpaired two-tailed *t*-test. n.s.; not significant.

B    Cells transfected with 20 ng pairs of Casp2(C320A) BiFC constructs (wild type or 3KR mutant) were treated as in (A) and subjected to GFP-Trap IP and analyzed by IB. endo.; endogenous.

C    Wild type or 3KR mutant pairs of Casp2pro-mVenus and Myc-Casp2pro constructs were co-expressed in HeLa cells. Cells were then treated with 20 μM cisplatin in the presence of 10 μM Q-VD(OMe)-OPh for 24 h, followed by GFP-Trap IP and salt washes at increasing stringencies (50, 150, 300, 500, and 750 mM NaCl).

D    Indicated amounts of catalytically active Casp2-mVenus constructs (wild type or 3KR mutant) were expressed in HEK293T cells for 24 h, and caspase-2 autoprocessing was assessed by IB.

E    Catalytically active Casp2-mVenus was expressed in HEK293T cells at sublethal levels for 24 h, followed by GFP-Trap IP. Immunoprecipitated Casp2-mVenus was eluted and subjected to VDVADase activity assay. *n* = 4 independent experiments (means + s.e.m.). *P* < 0.05 by unpaired two-tailed *t*-test (top). Expression of Casp2-mVenus was assessed by anti-GFP IB (bottom).

F    HeLa cells were transfected with catalytically active Casp2-mVenus for 24 h, then treated with 5 or 10 μM cisplatin for 24 h, and caspase-2 autoprocessing was assessed by IB.

G    Catalytically active Casp2-Myc (wild type or 3KR mutant) protein was expressed in HeLa cells for 24 h, followed by 5 μM cisplatin treatment for 24 h, and then analyzed by annexin V staining and flow cytometry. *n* = 4 independent experiments (means + s.e.m.). *P* < 0.05 by unpaired two-tailed *t*-test.

Source data are available online for this figure.

prodomain still dimerized in a TRAF2-dependent manner. This suggests that TRAF2 might bind to caspase-2 at more than one interface, with one being a canonical site (TIM) and the other atypical (prodomain). Indeed, the TIM may play an as yet undetermined functional role in activation of the full-length protein beyond dimerization. Alternatively, it may be that in the context of the full-length caspase-2, the TIM must bind TRAF2 to permit the prodomain-TRAF2 interaction (for example, due to conformational constraints of the full-length protein). Of course, it is also possible that TRAF2's interaction with the prodomain is indirect but still functionally critical.

TRAF2 can homo- or hetero-oligomerize and act as a scaffold to mediate the interaction of other proteins (Borghi *et al*, 2016). Thus, TRAF2 might be promoting caspase-2 activation as an adaptor by binding and inducing dimerization. TRAF2 has also been shown to act as a RING domain-dependent ubiquitin E3 ligase (Alvarez *et al*, 2010; Gonzalvez *et al*, 2012). In line with this, we found that caspase-2 BiFC dimers were ubiquitylated in response to caspase-2-activating stimuli and that TRAF2 can directly ubiquitylate caspase-2 (Fig 5A and E). When we mutated prominent ubiquitylation sites at K15, K152, and K153 (3KR) in the caspase-2 prodomain, we found that caspase-2 could still dimerize, but the complex was unstable and had reduced catalytic activity (Fig 7). This suggests that TRAF2 can promote caspase-2 activation as both an adaptor to initiate dimerization (as indicated by the loss of dimerization following TRAF2 knockdown or mutation of the caspase-2 TIM) and an E3 ubiquitin ligase to stabilize the dimer.

K152 and K153 were previously identified as part of a nuclear localization signal (NLS) in caspase-2 (Paroni *et al*, 2002; Baliga *et al*, 2003), and their mutation to alanine abolished nuclear import. We used arginine substitutions in this study as they maintain a longer, basic residue at the position of interest and thus are less disruptive than alanine mutations (Hodel *et al*, 2001; Kosugi *et al*, 2008). Similar to Paroni *et al* (2002), we found that mutation of K152 and K153 did not have a significant impact on caspase-2 activity induced by high overexpression. However, these mutations dampened activity when caspase-2 was expressed at more physiological levels (Fig 7D and E) and engaged by cisplatin (Fig 7F and G). Furthermore, TRAF2 overexpression induced caspase-2 foci formation in the cytoplasm and translocation to a detergent-insoluble fraction (Fig 6A and B), and this translocation was partially dampened by mutation of the caspase-2 ubiquitylation sites (Fig 6D). This suggests that both TRAF2 binding and ubiquitylation of these lysines changes caspase-2 localization in correlation with its activation, which is in line with previous findings that caspase-2 dimerization occurs in the cytoplasm (Bouchier-Hayes *et al*, 2009).

It is unclear how the ubiquitylation of caspase-2 promotes dimer stability. One potential mechanism is that ubiquitin-binding domain-containing proteins may serve as mediators of interaction that reinforce the dimer formation. Or perhaps the ubiquitylation of these lysines competes out other types of lysine-mediated modification (e.g., acetylation, methylation, neddylation, or SUMOylation) which might contribute to suppression of caspase-2 activation. Finally, caspase-2 ubiquitylation further enhances the TRAF2-caspase-2 interaction through an unknown mechanism (Fig 5H–J); perhaps this further recruitment is somehow critical for stabilization of the caspase-2 dimers or higher order structures. The caspase-2 dimer was previously shown to be stabilized by autoprocessing

(Baliga *et al*, 2004). It would be interesting to see in the future how ubiquitylation and cleavage cooperate to promote full proteolytic activity. A non-cleavable caspase-2 still retains some marginal proteolytic activity, although it is unable to effectively induce apoptosis. The 3KR mutation had a similar effect on caspase-2 (Fig 7D–G). Thus, regardless of how the two mechanisms interact, the full stabilization of caspase-2 through both mechanisms appears necessary for full proteolytic activity, enabling a cell to fully commit to apoptosis. We were intrigued to find that cellular extract from unstressed cells possesses a potent ability to deubiquitylate caspase-2 (Fig EV5B). This suggests that caspase-2 might be negatively regulated at the level of dimer stability through deubiquitylation to dampen enzyme activity and impair the initiation of apoptosis. Caspase-2 has been linked to non-apoptotic functions, such as promoting cell cycle arrest after cytokinesis failure (Fava *et al*, 2017). It is interesting to speculate that fine-tuning of caspase-2 ubiquitylation could determine which downstream pathway is activated, and whether caspase-2 acts as an inducer of apoptosis or a regulator of the cell cycle.

Non-degradative polyubiquitylation has previously been shown to promote both caspase-8 and caspase-1 activation, although its effect on dimer stability was not examined (Jin *et al*, 2009; Labbé *et al*, 2011). Furthermore, this polyubiquitylation of caspase-8 promotes its aggregation into punctate signaling structures in a detergent-insoluble fraction, a step required for its full activation (Jin *et al*, 2009); TRAF2-mediated caspase-2 ubiquitylation and cytoplasmic aggregation appear to play a similar role (Fig 6 and 7). Interestingly, TRAF2 can also ubiquitylate caspase-8 with K48-linked chains to target it for proteasomal degradation, thus shutting down death receptor-induced apoptosis (Gonzalvez *et al*, 2012). It would be intriguing to investigate how TRAF2 distinctly regulates these opposing effects on caspase-2 and caspase-8, or in other words, how it distinctly regulates intrinsic and extrinsic apoptosis pathways.

Taken together, we have shown that TRAF2 directly promotes caspase-2 dimerization and polyubiquitylation, which in turn enhances the recruitment of more TRAF2 to caspase-2, ultimately stabilizing the dimer complex and inducing full caspase-2 activation and apoptosis. Caspase-2 is emerging as an important regulator of tumorigenesis, metabolic syndrome, and neurodegenerative disease (Parsons *et al*, 2013; Puccini *et al*, 2013; Terry *et al*, 2015; Machado *et al*, 2016; Shalini *et al*, 2016; Zhao *et al*, 2016). Therefore, further investigation of this mechanism is needed to answer such questions as how TRAF2 is activated in response to caspase-2-activating stimuli, which linkage type of polyubiquitin chain is mediated by TRAF2, how ubiquitylation affects caspase-2 localization, and how this mechanism is negatively regulated. Such studies could yield insights into how to modulate caspase-2 as a therapeutic target.

# Materials and Methods

### Reagents and cell culture

GFP-Trap_A (cat. no. gta-20) and Myc-Trap_A (cat. no. yta-20) were from ChromoTek. Cisplatin (cat. no. P4394), doxycycline hydrochloride (cat. no. D3072), phosphatase inhibitor cocktail 2 (cat. no. P5726), phosphatase inhibitor cocktail 3 (cat. no. P0044),

N-ethylmaleimide (cat. no. E3876), polybrene (hexadimethrine bromide, cat. no. H9268), Anti-Flag M2 Affinity Gel (cat. no. A2220), anti-c-Myc agarose affinity gel (cat. no. A7470), phospho-creatine (cat. no. P7936), and GenElute Mammalian Total RNA Miniprep Kit (cat. no. RTN70) were from Sigma. Glutathione Sepharose 4B beads (cat. no. 17-0756-05) were from GE Health-care. Puromycin (cat. no. A11138-03) was from Thermo Fisher Scientific/Gibco. Hygromycin B (cat. no. 30-240-CR) was from Corning. Etoposide (cat. no. E-4488) and paclitaxel (cat. no. P-9600) were from LC Laboratories. Q-VD(OMe)-OPh (cat. no. A8165), TPCA-1 (cat. no. A4602), and IKK-16 (cat. no. B1586) were from APExBio. MG-132 (cat. no. BML-PI102) and sphin-gosine-1-phosphate (cat. no. BML-SL140-0001) were from Enzo Life Sciences. ZM447439 (cat. no. 13601) and dihydrocytochalasin B (DHCB) (cat. no. 20845) were from Cayman Chemical). Bradford Protein Assay Dye Reagent Concentrate (cat. no. 500-0006), iScript cDNA Synthesis Kit (cat. no. 170-8891), and iQ SYBR Green Super-mix (cat. no. 170-8882) were from Bio-Rad. QuikChange II Site-Directed Mutagenesis Kit (cat. no. 200524) was from Agilent Technologies. Tet System Approved FBS (cat. no. 63110) and In-Fusion HD Cloning Plus Kit (cat. no. 638909) were from Takara/Clontech. Pierce Silver Stain Kit for Mass Spectrometry (cat. no. 24600) and bismaleimidohexane (BMH) (cat. no. 22330) were from Thermo Scientific. AlexaFluor488-conjugated annexin V (cat. no. A13201) was from Thermo Fisher Scientific/Molecular Probes. Lipofectamine RNAiMAX Transfection Reagent (cat. no. 13778-150) was from Thermo Fisher Scientific/Invitrogen. Leupeptin (cat. no. 11529048001), aprotinin (cat. no. 11583794001), X-tremeGENE 9 DNA Transfection Reagent (cat. no. 06365809001), Anti-HA (3F10) Affinity Matrix beads (cat. no. 11815016001), and creatine kinase (cat. no. 10127566001) were from Roche Diagnostics. Polyethylen-imine (cat. no. 23966) was from Polysciences, Inc. Amylose resin (cat. no. E8021) was from New England Biolabs. Caspase-2 Substrate Ac-VDVAD-pNA (cat. no. 1072-1000) was from Bio-Vision. TNT T7 quick coupled transcription/translation systems (cat. no. L1170) was from Promega. Recombinant proteins: $His_6$-Ubiquitin E1 Enzyme (cat. no. E-304), UbcH5a (cat. no. E2-616), UbcH5b (cat. no. E2-622), UbcH5c (cat. no. E2-627), $His_6$-Ubc13 (cat. no. E2-660), $His_6$-Uev1a (cat. no. E2-662), ubiquitin (cat. no. U-100H), ubiquitin-aldehyde (cat. no. U-201), Mg-ATP solution (cat. no. B-20), and ubiquitin conjugation reaction buffer (cat. no. B-70) were from BostonBiochem.

HeLa, HEK293T, BT474, U-2OS, A549, and DAOY cells were obtained from Duke Cell Culture Facility (originally from ATCC). HeLa, HEK293T, U-2OS, A549, and DAOY cells were cultured in DMEM supplemented with 10% fetal bovine serum (Thermo Fisher Scientific/Gibco, cat. no. 16000-044). BT474 cells were cultured in RPMI1640 supplemented with 10% fetal bovine serum. Cells were grown in 37°C with 5% $CO_2$ and passaged every 2–3 days.

## Antibodies

Antibodies used in this study are as follows: caspase-2 mouse mono-clonal (G310-1248, cat. no. 551093, BD Biosciences), caspase-2 rat monoclonal (11B4, cat. no. MAB3507, EMD Millipore), actin rabbit polyclonal (I-19-R, cat. no. sc-1616-R, Santa Cruz Biotechnology), β-actin mouse monoclonal (cat. no. ab8224, Abcam), RAIDD mouse monoclonal (4B12, cat. no. M056-3, MBL), TRAF1 rabbit

monoclonal (45D3, cat. no. 4715, Cell Signaling Technology), TRAF2 mouse monoclonal (cat. no. 558890, BD Biosciences), TRAF2 rabbit polyclonal (C192, cat. no. 4724, Cell Signaling Tech-nology), TRAF2 rabbit polyclonal (cat. no. A303-460A, Bethyl Labo-ratories), TRAF3 rabbit polyclonal (cat. no. 18099-1-AP, ProteinTech), GFP mouse monoclonal (B-2, cat. no. sc-9996, Santa Cruz Biotechnology), GFP rabbit polyclonal (FL, cat. no. sc-8334, Santa Cruz Biotechnology), GFP rat monoclonal (3H9, cat. no. 3 h 9-100, ChromoTek), Flag mouse monoclonal (M2, cat. no. F1804, Sigma), Flag rabbit polyclonal (cat. no. PA1-984B, Thermo Scien-tific), Myc rabbit polyclonal (A-14, cat. no. sc-789, Santa Cruz Biotechnology), Myc mouse monoclonal (9B11, cat. no. 2276, Cell Signaling Technology), HA rabbit polyclonal (Y-11, cat. no. sc-805, Santa Cruz Biotechnology), caspase-3 rabbit monoclonal (8G10, cat. no. 9665, Cell Signaling Technology), active Bax mouse monoclonal (6A7, cat. no. 556467, BD Biosciences), Bax rabbit polyclonal (cat. no. 2772, Cell Signaling Technology), BID rabbit polyclonal (cat. no. 2002, Cell Signaling Technology), ubiquitin rabbit polyclonal (cat. no. 3933, Cell Signaling Technology), ubiquitin mouse monoclonal (P4D1, cat. no. sc-8017, Santa Cruz Biotechnology), MBP rabbit polyclonal (cat. no. ab9084, Abcam), PARP rabbit polyclonal (cat. no. 9542, Cell Signaling Technology), α-tubulin rabbit polyclonal (cat. no. ab4074, Abcam or cat. no. 2144, Cell Signaling Technol-ogy), Lamin B1 rabbit polyclonal (cat. no. ab16048, Abcam), caspase-1 rabbit monoclonal (D7F10, cat. no. 3866, Cell Signaling Technology), caspase-4 rabbit polyclonal (cat. no. 4450, Cell Signal-ing Technology), caspase-5 rabbit monoclonal (D3G4W, cat. no. 46680, Cell Signaling Technology), caspase-9 rabbit polyclonal (cat. no. 9502, Cell Signaling Technology), pSer32/36-IκBα mouse mono-clonal (5A5, cat. no. 9246, Cell Signaling Technology), IκBα rabbit polyclonal (C-21, cat. no. sc-371, Santa Cruz Biotechnology), pSer536-NF-κB p65 rabbit monoclonal (93H1, cat. no. 3033, Cell Signaling Technology), NF-κB p65 rabbit monoclonal (D14E12, cat. no. 8242, Cell Signaling Technology), NF-κB1 p105/p50 rabbit monoclonal (D7H5M, cat. no. 12540, Cell Signaling Technology), mono- and polyubiquitinylated conjugates mouse monoclonal (FK2, cat. no. BML-PW8810-0100, Enzo Life Sciences), K48-linkage-specific polyubiquitin rabbit polyclonal (cat. no. 4289, Cell Signaling Technology), and K63-linkage-specific polyubiquitin rabbit mono-clonal (D7A11, cat. no. 5621, Cell Signaling Technology).

## Plasmids

pBiFC-VN173 and pBiFC-VC155 were gifts from Dr. Chang-Deng Hu (Addgene plasmid # 22010 and # 22011, respectively) (Shyu et al, 2006). cDNA encoding caspase-2 prodomain (amino acid 1–169) or caspase-2 full length with C320A mutation (amino acid 1–452) was inserted into the multicloning site of pBiFC-VN173 or pBiFC-VC155 to generate plasmid encoding caspase-2 prodomain or caspase-2 (C320A) C-terminally fused with VN173 or VC155. For the inducible caspase-2 prodomain BiFC system, caspase-2 prodomain with VN173 or VC155 was inserted into the multicloning site I or II of pTRE-Tight-BI plasmid (Clontech, cat. no. 631068). Bidirectional Tet-responsive promoter in pTRE-Tight-BI enables simultaneous induction of both Casp2pro-VN173 and Casp2pro-VC155 in response to doxycycline washout.

mVenus-N1 was a gift from Dr. Michael Davidson (Addgene plasmid # 54640). cDNA sequence of caspase-2 full length or

prodomain was inserted into the multicloning site of mVenus-N1 to generate plasmid encoding C-terminal mVenus-tagged caspase-2 full length or prodomain. Caspase-2-mVenus without prodomain (Δ1–169) was created by PCR-based site-directed mutagenesis. Caspase-2 with C-terminal Myc or Flag expression constructs was generated in pcDNA3 by PCR-based subcloning. mCherry-H2A-10, which encodes C-terminally mCherry-tagged histone H2A, was a gift from Michael Davidson (Addgene plasmid # 55054).

pcDNA3-3xHA-TRAF1, Flag-TRAF2, Flag-TRAF3, Flag-TRAF5, and Flag-TRAF6 were generous gifts from Dr. Tomohisa Kato (Kyoto University, Japan) (Kato et al, 2008). pcDNA3-TRAF4 was a gift from Wafik El-Deiry (Addgene plasmid # 16375) (Sax & El-Deiry, 2003). TRAF7 cDNA originated from Dharmacon (MGC Human TRAF7 Sequence-Verified cDNA, Accession: BC024267, Clone ID: 5433246, cat. no. MHS6278-202832922). N-terminal 3xHA tag in the original TRAF1 expression construct was replaced with Flag tag by PCR-based subcloning. C-terminal Flag-tagged TRAF4 and N-terminal Flag-tagged TRAF7 constructs in pcDNA3 were created by PCR-based subcloning. Plasmids encoding N-terminal Myc-tagged TRAF2 wild type or TRAF2 RING deletion mutant were kindly provided by Dr. Vishva Dixit (Genentech, USA).

Site-directed mutagenesis was performed by PCR-based method using QuikChange II Site-Directed Mutagenesis Kit or In-Fusion HD Cloning Plus Kit.

### Generation of Tet system responsible caspase-2 BiFC expressing cell line

HeLa Tet-Off cells were transfected with both pTRE-Tight-BI/Casp2pro-VN173/Casp2pro-VC155 and Linear Hygromycin Marker (Clontech, cat. no. 631625). Cells were replated at low density and selected by 200 μg/ml Hygromycin B for 15 days until colony formed from single cell. Stably transfected clones were screened based on BiFC fluorescent signal, and Casp2pro BiFC protein expression was confirmed by immunoblot analysis.

### qPCR for Casp2pro BiFC expression

Casp2pro BiFC cells were cultured in the presence or absence of 1 μg/ml of doxycycline for 24 h, then treated with 20 μM cisplatin, 50 μM etoposide, or 100 nM paclitaxel in the presence of 10 μM Q-VD(OMe)-OPh. Cells were collected, and total RNA was extracted using GenElute Mammalian Total RNA Miniprep Kit. cDNA was synthesized using iScript cDNA Synthesis Kit and subjected to qPCR reaction using iQ SYBR Green Supermix. qPCR reaction was analyzed by CFX384 Real-Time System (Bio-Rad). All qPCR reactions were triplicated and repeated three times. Relative gene expression was converted using the $2^{-\Delta\Delta C_t}$ method with normalization by internal control GAPDH expression.

Primer sets used for qPCR were as below. Note that primers for Casp2-VN or Casp2-VC do not detect endogenous caspase-2.
Casp2pro-VN173 forward: 5′-ACTCCCTAGACAATAAAGATTCGG
Casp2pro-VN173 reverse: 5′-CTGAACTTGTGGCCGTTTAC
Casp2pro-VC155 forward: 5′-ACACTCCCTAGACAATAAAGATGG
Casp2pro-VC155 reverse: 5′-TCTGCTTGTCGTGGTTCATG
GAPDH forward: 5′-ACATCGCTCAGACACCATG
GAPDH reverse: 5′-ATGACAAGCTTCCCGTTCTC

### RNA interference and CRISPR/Cas9 gene editing

siRNA used in this study is as follows. siCtrl: AllStars Negative Control siRNA (cat. no. 1027281) was purchased from Qiagen. siTRAF1 and siRAIDD were purchased from Thermo Fisher Scientific/Ambion (siTRAF1: Silencer Select siRNA, ID s224753, siRAIDD: Silencer Select siRNA, ID s16654 (#1), s16655 (#2), s16656 (#3), and s225028 (#4)), and siTRAF2, siTRAF3, and siPIDD were purchased from Dharmacon (siTRAF2: cat. no. M-005198-00-0005 siGENOME SMARTpool, siTRAF3: cat. no. L-005252-00-0005 ON-TARGETplus SMARTpool, siPIDD: cat. no. LQ-004428-00-0002 ON-TARGETplus Set of 4). siRNA targeting caspase-2 was custom-designed, and the sequence is 5′-GCGAAUUGUUAGAACAUCUUCUGGA (CDS-targeting), or 5′-UGGAAGUAUUUGAGAGAGA (3′-UTR-targeting). siRNA transfection was performed following manufacturer's protocol of Lipofectamine RNAiMAX Transfection Reagent. 20 nM siRNA at final concentration was used.

Lentivirus encoding shRNA was generated using pLKO.1-based plasmids. Scramble shRNA (shNT) was a gift from David Sabatini (Addgene plasmid # 1864) (Sarbassov et al, 2005). shTRAF2 #1 (TRCN 0000004573), shTRAF2 #2 (TRCN0000004571), shCasp2 #1 (TRCN000 0003508), and shCasp2 #2 (TRCN0000003509) were obtained from Duke Functional Genomics Shared Resource (originally from Sigma). pLKO.1 plasmid and plasmids encoding gag, pol, tat, or VSV-G were transfected into HEK293T cells with polyethylenimine. Two days later, cell culture supernatant containing lentivirus was collected and filtered through 0.45-μm filter. Lentivirus was concentrated by ultracentrifugation at 65,000 g for 3 h at 4°C. Cells were transduced with the lentivirus in the presence of 8 μg/ml polybrene and selected with 2 μg/ml puromycin for at least 4 days before functional analysis.

Lentivirus encoding gRNA and Cas9 was generated using lentiCRISPR v2 plasmid: lentiCRISPR v2 was a gift from Dr. Feng Zhang (Addgene plasmid # 52961) (Sanjana et al, 2014). Target gRNA sequences for TRAF2 are 5′-CCTGCGGAGGACGTTTCTGC (#1) and 5′-ATATATGCCCTCGTGAACAC (#2) and were previously described (Shalem et al, 2014). Lentivirus production and transduction were carried out as above.

### Co-immunoprecipitation

Constructs were transfected into cells and cultured for 24–48 h. Cells were harvested by trypsinization and washed with PBS, then lysed in Co-IP lysis buffer A (0.4% Triton X-100, 40 mM HEPES (pH 7.4), 1 mM EDTA, 120 mM NaCl) or Co-IP lysis buffer B (0.5% Triton X-100, 20 mM Hepes (pH 7.4), 150 mM NaCl, 1.5 mM MgCl₂, 2 mM EGTA), supplemented with 10 μg/ml each aprotinin and leupeptin, 1 mM PMSF. Cell lysates were cleared by centrifugation at 4°C. Equal amounts of total protein between samples were incubated with the pre-washed appropriate affinity gel (either Anti-Flag M2 Affinity Gel, anti-c-Myc agarose affinity gel, or GFP-Trap_A) under constant rotation 4°C for several hours or overnight. Then, beads were washed four times with Co-IP lysis buffer, and bound proteins were eluted with SDS sample buffer. Samples were resolved by SDS–PAGE for immunoblot.

For the detection of endogenous interaction between caspase-2 and TRAF2, co-immunoprecipitation was carried out with rabbit anti-TRAF2 antibody (Bethyl Laboratories) and Protein A

Dynabeads, followed by immunoblot with rat anti-caspase-2 antibody (EMD Millipore) and mouse anti-TRAF2 antibody (BD Biosciences).

## Active Bax detection by anti-active Bax antibody 6A7

Cells were treated for 18 h with 20 μM cisplatin and then lysed with CHAPS lysis buffer (1% CHAPS, 150 mM NaCl, 10 mM Hepes (pH 7.4)). Anti-active Bax 6A7 antibody was prebound to protein G sepharose beads and then incubated with equal amounts of protein between samples for 1 h under constant rotation at 4°C. After incubation, sepharose beads were washed five times with CHAPS lysis buffer. Samples were resolved by SDS–PAGE for immunoblot.

## Microscopic or flow cytometric analysis of BiFC signal

HeLa Tet-Off/Casp2pro BiFC cells were plated on 35-mm glass bottom dishes (MatTek Corporation, P35G-1.5-14-C) for microscope or 12-well plate for flow cytometry at ~30% confluence for next day. Expression of Casp2pro BiFC was induced by washing with PBS and culturing cells in DMEM containing 10% Tet System Approved FBS for 24 h before treatment. Cells were treated with 20 μM cisplatin, 50 μM etoposide, or 100 nM paclitaxel in the presence of 10 μM Q-VD(OMe)-OPh for 24 h. Casp2pro BiFC signal was analyzed by FACScan Analyzer (BD Biosciences) or observed by Leica SP5 inverted confocal microscope with 40× objective lens. Acquired microscopic images were analyzed by Leica LAS AF software. For quantitation, Casp2pro BiFC signal was detected by flow cytometry using FACScan or FACSCalibur analyzers.

For detection of BiFC based on full-length caspase-2, HeLa cells were transiently transfected with pairs of plasmids encoding Casp2 (C320A)-VN173 and Casp2(C320A)-VC155. 24 h post-transfection, cells were treated with 20 μM cisplatin in the presence of 10 μM Q-VD(OMe)-OPh for 24 h. Cells were collected by trypsinization, and Casp2 BiFC signal was analyzed by FACScan Analyzer.

## BiFC dimer immunoprecipitation by GFP-Trap

Cells were trypsinized, and a portion of the cell suspension was analyzed by FACScan Analyzer or FACSCalibur for BiFC fluorescent signal. Cells were collected by centrifugation and lysed in lysis buffer (0.5% Triton X-100, 20 mM Hepes (pH 7.4), 150 mM NaCl, 1.5 mM MgCl$_2$, 2 mM EGTA, 2 mM DTT) or RIPA buffer (1% NP-40, 0.1% SDS, 0.1% sodium deoxycholate, 50 mM Tris (pH 7.4), 150 mM NaCl, 1 mM EDTA) supplemented with protease inhibitors and phosphatase inhibitors. GFP-Trap_A was pre-washed with lysis buffer, lysate was added, and the samples were incubated for 1–2 h under constant rotation at 4°C. GFP-Trap_A beads were washed with lysis buffer three times, and the immunoprecipitated protein was eluted in SDS sample buffer (2% SDS, 6% glycerol, 30 mM Tris (pH 6.8), 1% β-mercaptoethanol) by boiling for 10 min. Immunoprecipitated proteins were analyzed by immunoblot.

## Co-immunoprecipitation with BiFC dimer-specific capture and mass spectrometry analysis of the protein complex

HeLa Tet-Off/Casp2pro BiFC cells were plated at ~60% confluence for next day. Cells were thoroughly washed with PBS five times

to completely washout residual doxycycline and cultured in DMEM supplemented with Tet System Approved FBS for the induction of Casp2pro BiFC. For control cells, 1 μg/ml doxycycline was added to suppress Casp2pro BiFC expression, and 1 μg pair of pBiFC-VN173 and pBiFC-VC155 plasmids were transfected for overexpression of VN173 and VC155 proteins. After 24-h Casp2pro BiFC induction, cells were treated with DMSO, 20 μM cisplatin, 50 μM etoposide, or 100 nM paclitaxel in the presence of 10 μM Q-VD(OMe)-OPh. 24 h post-treatment, cells were collected by trypsinization and washed with cold PBS twice. Cell pellets were lysed in co-immunoprecipitation lysis buffer (0.5% Triton X-100, 20 mM Hepes (pH 7.4), 150 mM NaCl, 1.5 mM MgCl$_2$, 2 mM EGTA, 2 mM DTT) supplemented with protease inhibitors and phosphatase inhibitors. Cell lysates were cleared by centrifugation at 13,000 g for 15 min, and the supernatant was diluted with 1.5 × volume of dilution buffer (20 mM Hepes (pH 7.4), 400 mM NaCl, 1.5 mM MgCl$_2$, 2 mM EGTA, 2 mM DTT) to reduce Triton X-100 concentration to 0.2% and to increase NaCl concentration to 300 mM. Protein concentration was measured by Bradford assay, and 4.4 mg of total protein was used for co-immunoprecipitation with 13 μl slurry of GFP-Trap_A beads. At this step, control sample lysates from different treatments were mixed before GFP-Trap_A co-immunoprecipitation. The mixture of lysate and GFP-Trap_A was incubated under constant rotation at 4°C for 2 h. Then, GFP-Trap_A beads were washed seven times with wash buffer (1% Triton X-100, 20 mM Hepes (pH 7.4), 300 mM NaCl, 1.5 mM MgCl$_2$, 2 mM EGTA, 2 mM DTT) supplemented with protease inhibitors and phosphatase inhibitors, and three times with 50 mM ammonium bicarbonate. Samples were subjected to direct on-resin digestion and mass spectrometry for interaction proteomics at the Duke Proteomics and Metabolomics Core Facility.

## Apoptotic cell death detection by annexin V staining

Cells were plated on 6-well plate for about 40% confluence for next day. In some experiments, cells were transfected with plasmid encoding catalytically active caspase-2 and cultured for 24 h post-transfection. Then, cells were treated with cisplatin, etoposide, or paclitaxel. Medium containing floating cells and cells trypsinized from the culture dish were collected and washed with PBS. Washed cells were resuspended in 50 μl of annexin V-binding buffer (10 mM HEPES, 140 mM NaCl, and 2.5 mM CaCl$_2$, pH 7.4) supplemented with 2.5 μl of annexin V, Alexa Fluor 488 conjugate and incubated at room temperature for 15 min. Then, 200 μl of annexin V-binding buffer was added, and annexin V-positive cells were detected by flow cytometry using FACScan Analyzer or FACSCalibur.

## Caspase-2 localization analysis by microscope

HeLa cells were plated on 35-mm glass bottom dish at ~30% confluence for next day. Cells were transfected with plasmids encoding Casp2(C320A)-mVenus, Myc-TRAF2 (wild type or RING domain mutant), and H2A-mCherry, and cultured for 2 days post-transfection. mVenus and mCherry fluorescence was observed by Leica SP5 inverted confocal microscope with 63× objective lens. Acquired images were analyzed by Leica LAS AF software.

## Cell fractionation and protein extraction

Cells were collected, washed twice with PBS, and lysed in 0.1% NP-40 lysis buffer (0.1% NP-40, 10 mM Hepes (pH 7.4), 150 mM NaCl, 10% glycerol, 1 mM EDTA) supplemented with protease inhibitors, phosphatase inhibitors, and 10 mM N-ethylmaleimide, and left on ice for 15 min. Lysates were centrifuged at 15,000 $g$, 4°C for 10 min, and the supernatant was taken as 0.1% NP-40 soluble fraction. The residual pellet was lysed in RIPA buffer supplemented with protease inhibitors, phosphatase inhibitors, and 10 mM N-ethylmaleimide. The pellet suspension was sonicated to shear insoluble cell materials and vigorously mixed by vortex for 10 min at 4°C. Lysates were cleared by centrifugation at 15,000 $g$, 4°C for 15 min, and the supernatant was taken as 0.1% NP-40 insoluble fraction.

## Detection of caspase-2 ubiquitylation

Cells were transfected with plasmid encoding Casp2pro-mVenus (wild type or each lysine-to-arginine (KR) mutant) or mVenus-N1 as control. 24 h post-transfection, cells were collected and lysed in RIPA buffer supplemented with protease inhibitors, phosphatase inhibitors, and 10 mM N-ethylmaleimide. Cell debris was sheared by sonication, and lysates were cleared by centrifugation. Casp2pro-mVenus protein was captured by GFP-Trap_A under constant rotation at 4°C for 1 h. Beads were washed three times with harsh wash buffer (PBS containing 8 M urea and 0.2% SDS) to washout any other ubiquitylated protein associating with Casp2pro-mVenus. The samples were analyzed by SDS–PAGE following immunoblot with anti-ubiquitin antibody for the detection of ubiquitylated Casp2pro-mVenus.

For detection of endogenous caspase-2 ubiquitylation, cell lysates were denatured by boiling at 100°C in the presence of 1% SDS, followed by dilution with lysis buffer without SDS. Caspase-2 protein was immunoprecipitated with mouse anti-caspase-2 antibody (BD Biosciences) and Protein G Dynabeads. Ubiquitylation was analyzed by immunoblot with anti-ubiquitin antibody.

Cells transfected with HA-ubiquitin were lysed in RIPA buffer supplemented with protease inhibitors, phosphatase inhibitors, and 10 mM N-ethylmaleimide. Lysates were sonicated to shear cell debris and cleared by centrifugation. Anti-HA (3F10) Affinity Matrix beads were mixed with the lysate and rotated for 2 h at 4°C to pull down proteins modified with HA-ubiquitin. Beads were washed three times with harsh wash buffer. Captured proteins were eluted in SDS sample buffer by boiling for 10 min and analyzed by immunoblot.

## In vitro deubiquitylation assay

HeLa cells were transfected with plasmid encoding Casp2pro-mVenus and HA-ubiquitin, and cultured for 48 h post-transfection. Cells were collected and lysed in RIPA buffer supplemented with protease inhibitors, phosphatase inhibitors, and 10 mM N-ethylmaleimide. Lysates were sonicated to shear cell debris and cleared by centrifugation at 15,000 $g$, 4°C for 15 min. Protein concentration was measured by Bradford assay, and 13 mg of total protein was subjected to pull down with 5 μl slurry of GFP-Trap_A beads and incubated by constant rotation for 2.5 h at 4°C. GFP-Trap_A beads

were washed three times with wash buffer (1% Triton X-100, 20 mM Hepes (pH 7.4), 300 mM NaCl, 1.5 mM MgCl₂, 2 mM EGTA, 2 mM DTT) and once with PBS. Captured protein was eluted in 0.2 M glycine-HCl (pH 2.5) under constant vortex for 30 s at room temperature followed by immediate neutralization with 1 M Tris–HCl (pH 10.4). This step was repeated to increase elution efficiency. Eluted Casp2pro-mVenus with HA-ubiquitin chain was used as substrate for *in vitro* deubiquitylation assay. In parallel, untransfected HeLa cells were collected and resuspended in hypotonic buffer (20 mM Hepes-KOH (pH 7.5), 10 mM KCl, 1.5 mM MgCl₂, 1 mM EDTA, 1 mM DTT) supplemented with protease inhibitors. Cell suspension was kept on ice for 30 min and then homogenized with 100 strokes. Cell debris was sheared by sonication, and lysates were cleared by centrifugation at 15,000 $g$, 4°C for 30 min. 90 μl of the lysates was mixed with the eluted substrate (Casp2pro-mVenus with HA-ubiquitin chain) and recombinant MBP-TRAF2 or MBP in the presence of 20 μM MG132. Total reaction volume was adjusted to 100 μl. The reaction mixture was incubated at 37°C for 0, 0.5, 1.5, or 3.5 h, and then, reaction was stopped by adding SDS sample buffer and boiling for 10 min. Deubiquitylation of the substrate was monitored by immunoblot.

## Recombinant protein production

PCR-amplified TRAF2 cDNA was inserted into pMAL-c2X (New England Biolabs) to obtain a construct encoding N-terminal MBP-tagged TRAF2 (pMAL/MBP-TRAF2). Rosetta2(DE3)pLysS competent cells (EMD Millipore, cat. no. 71403) were transformed with pMAL/MBP-TRAF2 and cultured in LB medium at 37°C for overnight. 0.1 mM IPTG was added when OD₆₀₀ reached around 0.4, and the transformed cells were further cultured at 30°C for 6 h. Cells were collected by centrifugation and resuspended in Buffer A (20 mM Hepes-KOH (pH 7.5), 10 mM KCl, 1.5 mM MgCl₂, 1 mM EDTA, 1 mM EGTA, 1 mM DTT) supplemented with protease inhibitors. Cell debris was sheared by sonication, and lysates were cleared by centrifugation at 10,000 $g$ for 1 h. Recombinant MBP-TRAF2 protein was captured by glutathione sepharose beads at constant rotation at 4°C for 1 h. Beads were washed with Buffer A containing 1 M NaCl eight times and once with Buffer A. Sequential elution of the recombinant protein was performed with Buffer A containing 20 mM glutathione (pH 8.0). Elution fractions containing high amounts of MBP-TRAF2 were identified by SDS–PAGE and coomassie brilliant blue (CBB) staining, and selected fractions were dialyzed in Buffer A at 4°C overnight. Final purity and concentration of the recombinant protein was determined by SDS–PAGE and CBB staining.

## In vitro binding assay of ubiquitylated caspase-2 and recombinant MBP-TRAF2

HeLa cells were transfected with plasmid encoding Casp2pro-mVenus (wild type or 3KR mutant). Two days later, cells were collected and lysed in 1% NP-40 lysis buffer (1% NP-40, 150 mM NaCl, 50 mM Tris (pH 7.4), 1 mM EDTA) supplemented with protease inhibitors, phosphatase inhibitors, and 10 mM N-ethylmaleimide. Cell debris was sheared by sonication and then cleared by centrifugation. One μg of recombinant MBP-TRAF2 or MBP protein was added to the lysate and incubated on ice for 1 h. MBP-TRAF2

or MBP proteins were retrieved by 20 µl slurry of amylose resin at constant rotation for 1 h. Then, beads were washed four times with lysis buffer, and samples were analyzed by immunoblot.

### Salt titration assay to test the strength of caspase-2 dimer stability

HeLa cells were plated in 10-cm dishes and transfected with both 30 ng of plasmid encoding Casp2pro-mVenus and 70 ng of plasmid encoding Myc-Casp2pro (wild type or 3KR mutant) on next day. Cells were cultured for 24 h post-transfection and treated with 20 µM cisplatin in the presence of 10 µM Q-VD(OMe)-OPh in order to induce caspase-2 dimerization. After 24-h culture, cells were collected and lysed in lysis buffer (1% NP-40, 20 mM Hepes (pH 7.4), 50 mM NaCl, 2 mM EDTA) supplemented with protease inhibitors, phosphatase inhibitors, and 10 mM N-ethylmaleimide. Lysates were cleared by centrifugation, and high salt lysis buffer (1% NP-40, 20 mM Hepes (pH 7.4), 2 M NaCl, 2 mM EDTA) was added to increase the NaCl concentration up to 150, 300, 500, or 750 mM. Dimer of Casp2pro-mVenus and Myc-Casp2pro was immunoprecipitated by GFP-Trap_A under constant rotation at 4°C for 2 h. Beads were washed four times with lysis buffer with 50, 150, 300, 500, or 750 mM NaCl and once with lysis buffer with 50 mM NaCl. Samples were analyzed by immunoblot.

### *In vitro* ubiquitylation and ubiquitylation/binding assay

N-terminally Myc- or Flag-tagged TRAF2 (wild type or RING domain mutant) was overexpressed in HEK293T cells and cultured for 48 h post-transfection. Cells were collected and washed twice with PBS, and lysed in 1% NP-40 lysis buffer (1% NP-40, 50 mM NaCl, 50 mM Tris (pH7.4), 1 mM EDTA) supplemented with protease inhibitors and phosphatase inhibitors. Cell debris was shredded by sonication, and lysates were cleared by centrifugation at 15,000 *g*, for 15 min. 7 mg of total protein was subjected to pull down by Myc-Trap_A or anti-Flag affinity beads and rotated for 2–3 h at 4°C. Beads were washed once with 1% NP-40 lysis buffer with 300 mM NaCl, three times with 1% NP-40 lysis buffer with 50 mM NaCl, and once with PBS. The resultant purified recombinant TRAF2 protein on beads was used as ubiquitin E3 ligase for *in vitro* ubiquitylation assay. Casp2-Myc or Casp2-Flag recombinant protein was produced using TNT T7 quick coupled transcription/translation systems and used as substrate for the *in vitro* ubiquitylation assay. These recombinant TRAF2 and caspase-2 were mixed with 100 µM sphingosine-1-phosphate, 37 nM His$_6$-Ubiquitin E1 enzyme, 500 nM E2 enzymes (UbcH5a,b,c, His$_6$-Ubc13, and His$_6$-Uev1a), 47 µM ubiquitin, 2.5 µM ubiquitin-aldehyde, 20 mM phosphocreatine, 100 µg/ml creatine kinase, 20 µM MG132, and 10 mM Mg-ATP in Ubiquitin conjugation reaction buffer. Reaction volume was adjusted to 25 µl. Reactions were incubated at 37°C for 3 h. Ubiquitylation reaction was stopped by adding SDS sample buffer and boiling for 10 min.

For ubiquitylation/binding assay, Flag-tagged TRAF2 on anti-Flag beads and recombinant Casp2-Myc were subjected to *in vitro* ubiquitylation assay as described above. After 3-h reaction at 37°C, anti-Flag beads in the reaction were washed three times with wash buffer (1% Triton X-100, 300 mM NaCl, 20 mM Hepes (pH 7.4),

1.5 mM MgCl$_2$, 2 mM EGTA, 2 mM DTT). Proteins on the beads were eluted in SDS sample buffer by boiling for 10 min.

### VDVADase activity assay for caspase-2 purified from cells

HEK293T cells were plated in 10 cm dishes and transfected with plasmid encoding Casp2-mVenus (wild type, 3KR, C320A, or 3KR/C320A). Note that the amount of plasmid for transfection was 75 ng per dish which did not induce cell death 24 h post-transfection. Cells were collected and lysed in lysis buffer (0.5% Triton X-100, 20 mM Hepes (pH 7.4), 150 mM NaCl, 1.5 mM MgCl$_2$, 2 mM EGTA, 2 mM DTT) by incubating on ice for 30 min, and cleared by centrifugation at 15,000 *g*, 4°C. Casp2-mVenus was purified from cleared lysates with GFP-Trap_A under constant rotation at 4°C for 1 h. Beads were washed four times with lysis buffer and once with PBS. The immunoprecipitated Casp2-mVenus on beads was eluted with 0.2 M glycine-HCl (pH 2.5) under constant vortex at room temperature for 30 s following immediate neutralization with 1 M Tris–HCl (pH 10.4). This step was repeated to increase elution efficiency. The elution was mixed with caspase-2 substrate Ac-VDVAD-pNA in caspase assay buffer (50 mM Hepes (pH 7.4), 0.1 M NaCl, 0.1% CHAPS, 1 mM EDTA, 10% glycerol). The reaction was prepared in a 96-well plate, and Casp2-mVenus activity was measured as cleaved pNA signal at OD 405 nm under 37°C incubation up to 90 min using a Model 680 Microplate Reader (Bio-Rad). Equal amounts of Casp2-mVenus protein were added to each reaction, as confirmed by immunoblot.

### Statistical analysis

Data from flow cytometry-based experiments were shown as mean + s.e.m. Immunoblot results were analyzed and quantified by the ImageJ software. Each experiment was repeated at least three times, and the number of replicates (*n*) is indicated in the figure legend. Data were analyzed by unpaired two-tailed Student's *t*-test. $P < 0.05$ was considered as statistically significant.

**Expanded View** for this article is available online.

### Acknowledgements

This work was supported in part by National Institutes of Health Grants GM080333 (to S. K.) and 1F31CA183463, Ruth L. Kirschstein National Research Service Awards (to K. R. L.). We thank J. Will Thompson, Laura Dubois, Sarah Mabbett, Arthur Moseley, and the rest of the Proteomics and Metabolomics Core facility for their help in planning and analyzing the MS analysis. We thank Lisa Bouchier-Hayes (Baylor College of Medicine) for technical advice for caspase-2 BiFC experiments. We also thank Chih-Sheng Yang, Erika Segear Johnson, Joshua L. Andersen, Nai-Jia Huang, Bofu Huang, Kim Cocce, Liguo Zhang, and other past and present Kornbluth Laboratory members for insightful discussion.

### Author contributions

KM and SK conceived the project. KRL, ACR, and KM designed experiments. KM, ACR, KRL, and JW performed experiments. ACR and KM wrote the manuscript, with contributions from KRL, JW, and SK.

### Conflict of interest

The authors declare that they have no conflict of interest.

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
