## [Review Process File · The EMBO Journal]

Dimer-specific immunoprecipitation of active caspase-2 identifies TRAF proteins as novel activators

Alexander C. Robeson, Kelly R. Lindblom, Jeffrey Wojton, Sally Kornbluth, and Kenkyo Matsuura.

Review timeline:

Submission date:	3 rd April 2017
Editorial Decision:	3 rd May 2017
Revision received:	19 th February 2018
Editorial Decision:	5 th April 2018
Revision received:	30 th April 2018
Accepted:	7 th May 2018

Editor: Karin Dumstrei

Transaction Report:

1st Editorial Decision

3rd May 2017

Thank you for submitting your manuscript to The EMBO Journal. Your study has now been seen by three referees and their comments are provided below.

As you can see from the reports there is an interest in the findings. However, it is also clear that significant revisions are needed. The concerns raised are clearly outlined below and would have to be addressed in full in order to consider publication here. Importantly, further support for that caspase-2 is ubiquitinated at endogenous level is needed and extending the findings to an additional cell line would also strengthen the key conclusions.

Should you be willing to embark on a significant revision process then I am open to consider a revised manuscript. I should add that it is EMBO Journal policy to allow a single major round of revision only and that it is therefore important to address the raised concerns at this stage. So please consider your options carefully. Having said that I can offer to extend the revision time to 6 months.

It would be good if you could get back to me to let me know if you will be able to address the concerns and the timeline needed for to carry out the revisions.

When preparing your letter of response to the referees' comments, please bear in mind that this will form part of the Review Process File, and will therefore be available online to the community. For more details on our Transparent Editorial Process, please visit our website:

http://emboj.embojpress.org/about#Transparent_Process

Thank you for the opportunity to consider your work for publication. I look forward to your revision.

REFeree REPORTS.

Referee #1:

This is a highly interesting piece of work (re)identifying TRAF family proteins, in particular TRAF2, as interactors of Caspase-2. The authors provide evidence that this interaction may become relevant e.g. in response to DNA damage-inducing drugs, most prominently Cisplatin, or anti-mitotics, both implicated to activate Caspase-2.

The authors provide further evidence that Caspase-2 becomes ubiquitinated in a TRAF2-dependent manner which appears to stabilize Caspase-2 in its active (dimeric) form, generating enough proteolytic activity to kill HeLa cells. Cell death under these conditions appears to be dependent on the BCL2 family proteins BID and BAX/BAK, but could also engage others. Correlative evidence is provided that this process may also happen in another cell lines, BT474 breast cancer cells, that show similar resistance to drug-treatment when transfected with siRNAs targeting C2 or TRAF2. Yet, no evidence is provided that the protective effect of TRAF2 knockdown is mediated via ubiquitination of C2.

The initial approach chosen to identify novel C2 interactors is really elegant, as only active Caspase-2 (or at least dimerized C2) is enriched after drug-treatment and pulled down for mass spec based interrogation of binding partners. A potential problem regarding the selling strategy here is the fact that interaction of Caspase-2 with TRAF1 and TRAF2 has been reported a long time ago (Lamkanfi, 2005; cited by the authors). So, the title is certainly suboptimal, as this is clearly not a novel finding. Yet, this older study did only interrogate the role of C2 in NfκB signalling, not ubiquitination or cell death.

I think the study is generally of high quality and the conclusions are supported at least in part by the data shown. Clearly, TRAF2-dependent ubiquitination as a mode of regulation of Caspase2 activity is certainly novel and has certain implications for future work, given most recent developments in the Casp2 field. Yet, I believe there is a certain misconception regarding the biology behind C2 activation that prompted the authors to draw certain conclusions that are most likely not correct, as all this has (most likely) nothing to do with the DNA-damage response or cell death upon DNA damage as we know it.

Fava et al (G&D 01/2017) have shown recently that neither prolonged mitotic timing (induced by paclitaxel/taxol) nor DNA damage (e.g. Doxorubicin, CPT) triggers caspase-2 activation, but cell slippage / cytokinesis failure does. Looking at the IF pictures provided by the authors, there is a striking correlation of Venus expression and bi/multinucleation, suggesting that a substantial fraction of these cells fail cytokinesis after Cisplatin treatment (possibly also after Taxol). As HeLa cells lack a functional p53 response this may not be so surprising. This would also suggest that C2 kills such cells via the BCL2 network, a finding of interest, as Fava et al report mainly on C2-mediated cell cycle arrest in response to cytokinesis failure in p53 proficient cells. It would hence be very informative to see if treating these cells with DHCB or other compounds that trigger cytokinesis failure causes an increase in the percentage of BiFC positive cells. Using a p53 proficient system in comparison may be helpful to better discriminate these events.

The authors also suggest that neither RIPK1 nor PIDD are relevant here, as they have not been identified by MS. I think this argument is only partially valid. Regardless, RAIDD has been identified, and it would be of interest to interrogate if ubiquitination of C2 affects/facilitates RAIDD binding / localization, as the authors note that Ubi-C2 is more stable. Similarly, the effects of siRNA against RAIDD should be tested for impact on cell killing.

As Ubi-linkage-specific antibodies are available the authors should provide some insight into the type of chains formed on C2 - given the role of TRAF2 in NfκB signalling, K63 is most likely the mode of action and this should be confirmed/ruled out.

Finally, it would be really helpful to show (i) that the ubiquitination events can be detected on endogenous C2; (ii) in other cellular systems (at least when bringing in the C2 CARD) and (iii) to demonstrate that interaction of TRAF2 with the C2 CARD is specific over that of other CARDS, such as those found in Caspase-1, 4, 5 or 9.

Referee #2:

The manuscript by Robeson et al uses an elegant novel methodology to identify caspase-2 binding partners by adapting GFP-TRAP to pulldown refolded BiFC molecules that result from caspase-2 oligomerization. Using this approach they identify TRAF2 as a novel caspase-2 binding partner. The model proposed suggests that TRAF2, through ubiquitylation of caspase-2, serves to stabilize the caspase-2 dimer leading to increased activity. Not only does this paper use superior techniques to interrogate the caspase-2 pathway, it provides much needed insight into the mechanism of caspase-2 regulation. In general, the experiments are thoughtfully executed, well controlled and largely support the conclusions.

However, I do have a few concerns that if addressed would further strengthen this interesting study.

Major concerns

1. Figure 2 shows the interaction between caspase-2 and TRAF2. However, all the experiments are based on overexpression of one or both of the partners. Given how central this interaction is to the conclusions of the paper the authors should endeavor to show that the interaction can be detected between endogenous caspase-2 and endogenous TRAF2 (following cisplatin treatment for example) or provide an explanation as to why such an interaction cannot be detected. I could imagine, for example, that, once caspase-2 is cleaved, that the interaction may be difficult to detect but this needs to be formally addressed in the manuscript.

2. The major confusing element of the pathway is the fact that the binding motif for TRAF2 is located in the catalytic domain of caspase-2, while the interaction was identified with the prodomain of caspase-2 that does not encompass the TIM region. The authors do acknowledge this discrepancy in the discussion and postulate that there could be two binding sites for caspase-2 (the TIM and the CARD) or that the binding is indirect. However, I do not think this is sufficient, and some binding studies should be included to address this further. For example, does TRAF2 bind to a version of caspase-2 lacking the prodomain with more or less efficiency? Alternatively, given that caspase-2 was pulled down in the initial screen, is all this mediated by endogenous caspase-2 that complexes to the exogenously expressed caspase-2? If this is the case TRAF2 binding to caspase-2 pro would not be detected in cells that lack endogenous caspase-2.

3. While the novel interaction of caspase-2 and TRAF2 is interesting, the overall model and mechanism is not very clear. Is this a novel activation platform where TRAF2 promotes recruitment and activation of caspase-2, or is TRAF2 recruited to the preformed complex to stabilize the active enzyme complex. RAIDD is also pulled down in the initial IP but, curiously, this is not touched on again. Investigating if TRAF2-dependent caspase-2 activation is RAIDD-dependent or -independent would provide important insights into the nature of this complex.

4. In figure 7F and G the authors determine how expression of Casp2-mVenus wild type or 3KR mutant affects cisplatin-induced caspase-2 cleavage and apoptosis. Firstly, it is unclear in F if the cleavage fragments coming up at 35 and 15kDa represent cleavage of the endogenous or Venus-tagged caspase-2 (the same concern applies to 7D). Secondly, the differences in cleavage in F and apoptosis in G are quite small (albeit significant in the latter case). It is possible that the endogenous caspase-2 may be overriding the inhibitory effects of caspase-2 3KR. If these experiments were done in a caspase-2 null environment (by siRNA or other appropriate method), the authors would be able to more precisely determine the full extent of inhibition mediated by the 3KR mutant.

Minor concerns

None

Additional non-essential suggestions

I'm curious how the TRAF2-mediated stabilization of the caspase-2 dimer interplays with caspase-2 autocleavage that follows dimerization. Cleavage is considered to stabilize the active enzyme, so does TRAF 2 facilitate this or is it an independent stabilization mechanism. Would TRAF2's ability to stabilize the dimer be altered if autocleavage was blocked by mutating the cleavage sites? This is a little outside the scope of the initial study but inclusion of this in the discussion may be warranted.

Referee #3:

In this manuscript the authors aim to unveil a role for ubiquitination in the activation of caspase-2. Despite being cloned as one of the first mammalian caspases, the molecular machinery driving the activation of caspase-2 remains unclear. Thus the manuscript addresses a pertinent issue and therefore the story on caspase-2 biology is of potential interest to a broad readership. However, the manuscript suffers from serious technical concerns, over relying on "overexpression and/or nearly endogenous expression" experiments including fragmented caspase-2 coupled to tags. The fact that TRAF2-caspase-2 interaction has been demonstrated before (in the context of NfκB activation and the authors didn't address if these results are reproducible in their settings and if it has any role here) diminishes the enthusiasm to support publication the publication of this manuscript in EMBOJ. Thus the story might be appropriate for a specialized journal if the authors could convincingly address the following concerns.

- 1) One of the major concerns is the lack of convincing data demonstrating the ubiquitination of caspase-2 at endogenous levels. Most of the experiments are performed with "nearly endogenous" expression of BIFC-CARD domain (its claimed as caspase-2 in the manuscript in many places) in tumour cell lines. I'm surprised that the authors have not attempted well established techniques like "in situ trapping of caspases" with biotin-VAD which has been very well shown by several labs to precipitate the proximal caspases including caspase-2 (under both PIDD dependent and independent conditions). Ideally the authors should precipitate the dimerized caspase-2 with Biotin-VAD upon appropriate stimuli and check for ubiquitination at endogenous levels.
- 2) The authors failed to convincingly demonstrate that dimerized caspases-2 at endogenous levels is indeed ubiquitinated in a TRAF2 dependent manner. A direct role for ubiquitination in the dimerization /stabilisation of full-length caspase-2 is lacking. Gel filtration/ cross linking experiments could perhaps be employed (and gel filtration has been routinely employed by groups working on caspases-2 before to demonstrate caspases-2 oligomerisation in response to apoptotic stimuli) to demonstrate if dimerized caspase-2 in response to activating stimuli is indeed ubiquitinated in a Traf2 dependent manner. One should attempt the endogenous gel filtration by growing and lysing cells at 37°C and not by shifting the cell lysates from 4°C to 37°C (where caspases-2 can artificially get into high order multimers in a NaCl/KCl dependent manner).
- 3) TRAF2 /1 has been shown to be interacting with caspases-2 already and this reference is cited only in the discussion. But it should be cited already in page 4, when the authors make the claim that "we discover that caspases-2 dimers interact with TRAFs". As TRAF2 knock out cells are readily available the authors could test if caspase-2 activation is impaired in these cells at endogenous levels and if caspase-2 ub'n is impaired in response to cell death stimuli that activates caspase-2. Further, one can check if these cells are resistant to cisplatin and if this is dependent on caspase-2 activation. Again here one can test for dimerization of caspases-2 (biotin-vad) followed by processing or by cleavage of VDVAD-fmk. Here again, caspase-2 knock out cells could serve as a control to confirm the specificity of VDVAD-fmk cleavage.
- 4) The data on ubiquitination is not robust. To test in vivo ubiquitination of endogenous caspase-2 one should employ a denaturing/renaturing IP. Else TUBEs could also be employed. Ideally, one should precipitate Ub'nd proteins upon cisplatin stimulation and check for caspases-2 in TRAF2 knock out cells and in TRAF2 knock out cells complimented with wild type and RING deficient mutants. The authors can check for caspase-2 processing and cell death induction in the same assays. This will also infer if the caspases-2/TRAF-2 required for mediating cisplatin-mediated cell death in a cell type dependent manner.
- 5) It is surprising that the authors employed primarily the CARD -BIFC for most of the experiments and that's probably the reason why most of the Ubiquitination sites are identified in the N-terminus. These experiments should at the least be complimented with full-length catalytically dead caspase-2 mutants. Ideally, the authors should employ a Lys-Gly-GLY approach or other mass spec based approaches in an unbiased manner and look for ubiquitination sites on the entire caspase-2 protein precipitated from cells upon stimulation with DNA damaging agents. Or this can also be attempted on recombinant caspases-2 subjected to ubiquitination with TRAF-2. The identified sites could then be verified in vivo.
- 6) The requirement of caspase-2 in DNA damage-mediated apoptosis is controversial and may be

cell type dependent. While the authors tend to claim that they are looking at a scenario where caspases-2 activation is independent of PIDD and RAIDD, further experiments employing loss of function studies are needed to substantiate their claims here. Along these lines, it is not clear if caspases-3/7 is required for the activation of caspases-2 under these settings. If we go with published literature, one may need effector caspases for activation and /or processing of caspase-2 which also questions the role for Ub'n here. It is indeed possible that caspases-2 is activated in the absence of caspases-3/7 in their settings as a proximal caspase - But this needs to be demonstrated with employing caspases-3/7 double knock out cells.

7) Does TRAF2 overexpression stimulate caspases-2 activation in a RING dependent manner?

8) What kind of ubiquitin chains is synthesized on caspase-2 by TRAF2? Again here, mass spec/TUBE based approaches could shed some light which will enhance the mechanistic insights provided.

9) It's also not clear what is the functional significance of the translocation/shift of caspase-2 to detergent insoluble fraction. Again in Figure 6E, there is no evidence presented to demonstrate if the endogenous caspases-2 is ubiquitinated in the insoluble fraction.

10) In Figure 2, there is no evidence presented to claim that the interaction between TRAF2 and caspases-2 is induced upon cisplatin stimulation at endogenous levels.

11) In figure 3F: The authors should check for caspases-2 processing. Though cleavage is a secondary event and not necessarily an indication of activation of proximal caspases like caspases-2, a block in processing of caspase-2 could possibly support an upstream obligatory role for TRAF2 in caspases-2 activation. Further, is caspase-9 activation is also blocked?

12) As TRAF-2/Caspase-2 complex is shown to regulate NfKB, there is no evidence presented to argue that this pathway is not activated in response to the stimuli employed.

13) Figure 4: Are the TAQEM mutations impair the activation and processing of caspase-2 directly?

14) Figure 6A: The "puncta" needs to be quantified from multiple fields and from multiple experiments. As of now this figure is not so informative.

15) Figure 6D: Though overexpressed/ expressed to near endogenous levels, is the caspases-2(C320A)-venus ubiquitinated in the insoluble fraction?

1st Revision - authors' response

19th February 2018

We are grateful for the referees' comments. I believe we have been able to address the comments of the referees and hope you will now consider our manuscript suitable for publication in The EMBO Journal. Please find our point-by-point responses to the reviews below.

Referee #1:

This is a highly interesting piece of work (re)identifying TRAF family proteins, in particular TRAF2, as interactors of Caspase-2. The authors provide evidence that this interaction may become relevant e.g. in response to DNA damage-inducing drugs, most prominently Cisplatin, or anti-mitotics, both implicated to activate Caspase-2.

The authors provide further evidence that Caspase-2 becomes ubiquitinated in a TRAF2-dependent manner which appears to stabilize Caspase-2 in its active (dimeric) form, generating enough proteolytic activity to kill HeLa cells. Cell death under these conditions appears to be dependent on the BCL2 family proteins BID and BAX/BAK, but could also engage others. Correlative evidence is provided that this process may also happen in another cell lines, BT474 breast cancer cells, that

show similar resistance to drug-treatment when transfected with siRNAs targeting C2 or TRAF2. Yet, no evidence is provided that the protective effect of TRAF2 knockdown is mediated via ubiquitination of C2.

We believe that the following results support that the protective effect of TRAF2 knockdown is mediated via ubiquitylation of caspase-2. Firstly, TRAF2 knockdown significantly reduced ubiquitylation of caspase-2 (Figure 5C). Secondly, overexpression of TRAF2, but not RING domain mutant, increased ubiquitylation of caspase-2 (Figure 5D) and induced cell death in a caspase-2-dependent manner (Appendix Figures S7B and S7C). Importantly, the residual apoptosis-inducing activity of the TRAF2 RING domain mutant was not as affected by caspase-2 knockdown as that of the WT TRAF2 was, suggesting that the pro-apoptotic effects of TRAF2 overexpression that are exerted via caspase-2 are at least in part dependent on the ubiquitylating activity of TRAF2. Thirdly, the ubiquitylation-deficient 3KR mutant of caspase-2 exhibited less pro-apoptotic activity than wild type caspase-2. We therefore conclude that TRAF2-mediated ubiquitylation enhances caspase-2 activation and subsequent apoptosis. These results may not be direct evidence, and they don't exclude the possibility that TRAF2 could have a pro-apoptotic function other than positively regulating the caspase-2-mediated pathway through ubiquitylation.

Directly testing the significance of caspase-2 ubiquitylation is complicated by the finding that ubiquitylation is not the only manner by which TRAF2 activates caspase-2. This is demonstrated by the findings that TRAF2 knockdown blocks caspase-2 dimerization (Figures 3A and 3B), while elimination of the caspase-2 ubiquitylation sites (3KR) does not block initial formation of a dimer but does reduce the dimer's stability and activity (Figures 7A-G). Thus, it is difficult to look at caspase-2 ubiquitylation in isolation. If a caspase-2-specific deubiquitylating enzyme (hinted at in Appendix Figure S5B) is identified in the future, inhibiting or overexpressing that enzyme could allow us to study caspase-2 ubiquitylation in isolation. Unfortunately, that is beyond the scope of this manuscript.

The initial approach chosen to identify novel C2 interactors is really elegant, as only active Caspase-2 (or at least dimerized C2) is enriched after drug-treatment and pulled down for mass spec based interrogation of binding partners. A potential problem regarding the screening strategy here is the fact that interaction of Caspase-2 with TRAF1 and TRAF2 has been reported a long time ago (Lamkanfi, 2005; cited by the authors). So, the title is certainly suboptimal, as this is clearly not a novel finding. Yet, this older study did only interrogate the role of C2 in NfκB signalling, not ubiquitination or cell death.

We considered changing the title, but since the title refers to the novel role of TRAF2 in caspase-2 activation and not the novelty of the interaction, we decided to leave this as is. We did, however, move the reference to Lamkanfi earlier in the manuscript to make clear from the outset that this was a previously reported interaction.

I think the study is generally of high quality and the conclusions are supported at least in part by the data shown. Clearly, TRAF2-dependent ubiquitination as a mode of regulation of Caspase2 activity is certainly novel and has certain implications for future work, given most recent developments in the Casp2 field. Yet, I believe there is a certain misconception regarding the biology behind C2 activation that prompted the authors to draw certain conclusions that are most likely not correct, as all this has (most likely) nothing to do with the DNA-damage response or cell death upon DNA damage as we know it.

Fava et al (G&D 01/2017) have shown recently that neither prolonged mitotic timing (induced by paclitaxel/taxol) nor DNA damage (e.g. Doxorubicin, CPT) triggers caspase-2 activation, but cell slippage / cytokinesis failure does. Looking at the IF pictures provided by the authors, there is a striking correlation of Venus expression and bi/multinucleation, suggesting that a substantial fraction of these cells fail cytokinesis after Cisplatin treatment (possibly also after Taxol). As HeLa cells lack a functional p53 response this may not be so surprising. This would also suggest that C2 kills such cells via the BCL2 network, a finding of interest, as Fava et al report mainly on C2-mediated cell cycle arrest in response to cytokinesis failure in p53 proficient cells. It would hence be very informative to see if treating these cells with DHCB or other compounds that trigger cytokinesis failure causes an increase in the percentage of BiFC positive cells. Using a p53 proficient system in comparison may be helpful to better discriminate these events.

We appreciate the referee's thoughtful comments and suggestion. The Fava et al. paper (Fava *et al*, 2017) made an excellent case for the importance of caspase-2 in regulating the cell cycle after cell slippage or cytokinesis failure. We thus wanted to see how this finding fit in with our own. First, we transiently expressed caspase-2 BiFC constructs to examine caspase-2 activation by ZM447439 or DHCB treatment, both of which trigger cytokinesis failure and were used in the study by Fava et al. (Appendix Figures S1C and S1D). Consistent with their study, ZM447439 treatment of p53-proficient cell lines A549 and U-2OS induced a caspase-2 BiFC signal, indicating caspase-2 activation, to levels comparable with cisplatin treatment (Appendix Figure S1C). This seems to confirm that cytokinesis failure triggers caspase-2 dimerization and activation. However, in HeLa cells, ZM447439 treatment induced a significantly lower percentage of caspase-2 BiFC positive cells than

cisplatin treatment. Similarly, DHCB was a poor inducer of caspase-2 activation compared to cisplatin in HeLa Tet-Off Casp2pro BiFC cells (Appendix Figure S1D).

Fava et al also reported that caspase-2-mediated cell cycle arrest in p53-proficient cells depends on the PIDDosome complex. However, we found that cisplatin-induced caspase-2 activation in HeLa cells occurred independently of both RAIDD and PIDD (Appendix Figures S2B-F) (please see the next paragraph and our response to Referee #3 - comment #6). These results indicate that the DNA damage-induced caspase-2 activation mechanism and downstream pathway could be different in p53-deficient cancer cells compared with p53-proficient cells. We speculate that cytokinesis failure-induced caspase-2 activation by the PIDDosome is dependent on p53, potentially by p53-mediated PIDD induction as reported previously (Baptiste-Okoh *et al*, 2008; Lin *et al*, 2000; Oliver *et al*, 2011).

The authors also suggest that neither RIPK1 nor PIDD are relevant here, as they have not been identified by MS. I think this argument is only partially valid. Regardless, RAIDD has been indentified, and it would be of interest to interrogate if ubiquitination of C2 affects/facilitates RAIDD binding / localization, as the authors note that Ubi-C2 is more stable. Similarly, the effects of siRNA against RAIDD should be tested for impact on cell killing.

We have now examined the ability of the wild-type or ubiquitylation-deficient 3KR mutant of caspase-2 to bind RAIDD by co-immunoprecipitation. The 3KR mutant showed slightly weaker binding to RAIDD compared with wild-type caspase-2. However, the difference was subtle, and ubiquitylation of caspase-2 may not have a significant effect on RAIDD binding. The result is shown in Figure R for the reviewer's consideration (please see the end of this letter), but we have elected not to include this in the final manuscript as it doesn't add significantly to the overall findings.

We examined the effect of RAIDD knockdown on cisplatin-induced cell death and caspase-2 dimerization in HeLa cells. Three different sequences of siRNA were used for each assay, but none of them had an effect. The results are shown in Appendix Figures S2D-F. Taken together, RAIDD seems to not be involved in the activation of caspase-2 in our setting.

As Ubi-linkage-specific antibodies are available the authors should provide some insight into the type of chains formed on C2 - given the role of TRAF2 in NfkB signalling, K63 is most likely the mode of action and this should be

confirmed/ruled out.

We performed caspase-2 ubiquitylation assays with ubiquitin linkage-specific antibodies, as the referee suggested. mVenus tagged caspase-2 prodomain was expressed in HEK293T cells, then immunoprecipitated by GFP-Trap and blotted with several linkage-specific antibodies. The result is in Appendix Figure S5D. Consistent with our results in Figure 5F and 5G, anti-ubiquitin FK2 antibody (which reacts with K29-, K48-, and K63-linked ubiquitin chains) detected polyubiquitylated caspase-2. Interestingly, both K63- and K48-linkage specific anti-ubiquitin antibodies reacted with polyubiquitylated caspase-2. Of note, all of these polyubiquitylation events were disrupted by the 3KR mutation. This caspase-2 polyubiquitylation pattern might be indicative of branched polyubiquitin chains with mixed linkages, which have recently emerged as important factors in cellular processes like NF- κ B signaling and mitosis (Ohtake *et al*, 2016; Yau *et al*, 2017). Alternatively, each lysine (K15, K152, and K153) might undergo a different homotypic linkage of polyubiquitylation. Regardless, further experiments are required to distinguish the relevant polyubiquitin chain linkage types and how they regulate caspase-2.

Finally, it would be really helpful to show (i) that the ubiquitination events can be detected on endogenous C2; (ii) in other cellular systems (at least when bringing in the C2 CARD); (iii) to demonstrate that interaction of TRAF2 with the C2 CARD is specific over that of other CARDS, such as those found in Caspase-1, 4, 5 or 9.

(i) We examined endogenous caspase-2 ubiquitylation using a denaturing/renaturing immunoprecipitation assay. Specifically, the samples were boiled in SDS buffer, then diluted in IP Buffer and subjected to immunoprecipitation. While the ubiquitylation signal was weaker than that observed with transfected caspase-2, endogenous caspase-2 ubiquitylation was clearly increased after cisplatin treatment. The result is now in Figure 5B.

(ii) We examined the ubiquitylation status of the overexpressed caspase-2 prodomain in several additional cell lines – U-2OS, A549, and DAOY – and observed a signal similar to HeLa and HEK293T cells, though the signal was weaker in A549 and DAOY due to a lower transfection efficiency. The result is in Appendix Figure S5A.

(iii) We thank the referee for the intriguing suggestion. We tested whether the other CARD domain-containing caspases interact with TRAF2 upon its overexpression. Interestingly, caspase-9 was found to be another interactor of TRAF2 while caspase-1, -4, and -5 showed no interaction. Like caspase-2, caspase-9 is involved in the intrinsic apoptosis pathway, and it is possible that TRAF2 could regulate intrinsic apoptosis through

both of these caspases. The result is in Appendix Figure S2G. Of note, we showed that TRAF2 knockdown blocked DNA damage-induced intrinsic apoptosis upstream of the mitochondria, and therefore upstream of caspase-9 activation (Figures 3H and 3I).

Referee #2:

The manuscript by Robeson et al uses an elegant novel methodology to identify caspase-2 binding partners by adapting GFP-TRAP to pulldown refolded BiFC molecules that result from caspase-2 oligomerization. Using this approach they identify TRAF2 as a novel caspase-2 binding partner. The model proposed suggests that TRAF2, through ubiquitylation of caspase-2, serves to stabilize the caspase-2 dimer leading to increased activity. Not only does this paper use superior techniques to interrogate the caspase-2 pathway, it provides much needed insight into the mechanism of caspase-2 regulation. In general, the experiments are thoughtfully executed, well controlled and largely support the conclusions. However, I do have a few concerns that if addressed would further strengthen this interesting study.

Major concerns

1. Figure 2 shows the interaction between caspase-2 and TRAF2. However, all the experiments are based on overexpression of one or both of the partners. Given how central this interaction is to the conclusions of the paper the authors should endeavor to show that the interaction can be detected between endogenous caspase-2 and endogenous TRAF2 (following cisplatin treatment for example) or provide an explanation as to why such an interaction cannot be detected. I could imagine, for example, that, once caspase-2 is cleaved, that the interaction may be difficult to detect but this needs to be formally addressed in the manuscript.

We examined the interaction of endogenous caspase-2 and TRAF2 following cisplatin treatment. Per the referee's suggestion, the cells were cultured with the pan-caspase inhibitor Q-VD(OMe)-OPh to prevent caspase-2 cleavage, which might have attenuated the interaction. The result is in Figure 2G. Although some caspase-2 bound non-specifically to control IgG (or protein G resin), binding between endogenous TRAF2 and caspase-2 was clearly increased in a time-dependent manner with cisplatin treatment.

2. The major confusing element of the pathway is the fact that the binding motif for TRAF2 is located in the catalytic domain of caspase-2, while the interaction was identified with the prodomain of caspase-2 that does not encompass the TIM

region. The authors do acknowledge this discrepancy in the discussion and postulate that there could be two binding sites for caspase-2 (the TIM and the CARD) or that the binding is indirect. However, I do not think this is sufficient, and some binding studies should be included to address this further. For example, does TRAF2 bind to a version of caspase-2 lacking the prodomain with more or less efficiency? Alternatively, given that caspase-2 was pulled down in the initial screen, is all this mediated by endogenous caspase-2 that complexes to the exogenously expressed caspase-2? If this is the case TRAF2 binding to caspase-2 pro would not be detected in cells that lack endogenous caspase-2.

We agree with the referee that the binding mechanism was unclear, and appreciate the thoughtful suggestions. We examined a series of caspase-2 deletion mutants for TRAF2 binding: full-length, prodomain, or Δ 1-169 (no prodomain). The result is in Appendix Figure S4B. To exclude the possibility that endogenous caspase-2 might bridge the interaction between exogenous caspase-2 and TRAF2, endogenous caspase-2 was depleted by caspase-2 3'UTR-targeting siRNA. Interestingly, the caspase-2 prodomain could interact with TRAF2 even without endogenous caspase-2, although the binding was slightly weaker than full-length caspase-2. In contrast, the Δ 1-169 mutant showed much weaker binding compared with full-length or prodomain. This result, in combination with the TRAF interacting motif (TIM) mutant data (Figure 4B), indicates that both the prodomain and the TIM contribute to TRAF2 binding. Since mutation of either domain in the context of the full-length caspase-2 impedes binding, we conclude that both regions are necessary for optimal binding of TRAF2 in the context of the full-length caspase-2 and for optimal biological activity. It would be interesting to further investigate the role of these two binding motifs in future studies for a clearer understanding of caspase-2 activation.

3. While the novel interaction of caspase-2 and TRAF2 is interesting, the overall model and mechanism is not very clear. Is this a novel activation platform where TRAF2 promotes recruitment and activation of caspase-2, or is TRAF2 recruited to the preformed complex to stabilize the active enzyme complex. RAIDD is also pulled down in the initial IP but, curiously, this is not touched on again. Investigating if TRAF2-dependent caspase-2 activation is RAIDD-dependent or -independent would provide important insights into the nature of this complex.

Since TRAF2 is required for caspase-2 dimerization, as shown by caspase-2 BiFC (Figures 3A and 3B), it seems that TRAF2 initiates caspase-2 activation. However, TRAF2 functions as more than an adaptor, considering its role in stabilizing the active caspase-2 complex through ubiquitylation, which leads to further TRAF2 recruitment. Further analysis is required to clarify the relationship between TRAF recruitment

and caspase-2 activation. As for RAIDD, we examined its involvement in cisplatin-induced caspase-2 activation and apoptosis and found that it is dispensable in this context (please see our reply to the comment of referee #1 and Appendix Figures S2D-F).

4. In figure 7F and G the authors determine how expression of Casp2-mVenus wild type or 3KR mutant affects cisplatin-induced caspase-2 cleavage and apoptosis. Firstly, it is unclear in F if the cleavage fragments coming up at 35 and 15kDa represent cleavage of the endogenous or Venus-tagged caspase-2 (the same concern applies to 7D). Secondly, the differences in cleavage in F and apoptosis in G are quite small (albeit significant in the latter case). It is possible that the endogenous caspase-2 may be overriding the inhibitory effects of caspase-2 3KR. If these experiments were done in a caspase-2 null environment (by siRNA or other appropriate method), the authors would be able to more precisely determine the full extent of inhibition mediated by the 3KR mutant.

We appreciate the referee's concerns regarding potential complications from endogenous caspase-2. In regard to Figure 7D, the appearance of the cleavage fragments correlates well with the amount of transfected mVenus-tagged caspase-2, and we see a consistent pattern in Figure 7F. This leads us to believe that the cleavage fragments are derived from exogenous caspase-2. Furthermore, we believe that two factors suppress the difference between caspase-2 wild type and 3KR mutant cleavage observed in Figure 7F: (i) the 3KR mutant is expressed at higher levels, as seen in the absence of cisplatin (lanes 1 and 4) and (ii) cleaved caspase-2 is likely lost in the terminal stage of apoptosis because cells with processed caspase-2 are dying.

However, we agreed with the referee that it could be helpful to conduct the experiments in a caspase-2 null environment – we therefore examined the effect of the 3KR mutation in HeLa shCasp2 cells using shCasp2-resistant caspase-2 constructs. Interestingly, this did not improve the result – both caspase-2 cleavage and apoptosis were similar to what we saw in Figure 7F and 7G. The results are in Appendix Figure S7A. A possible reason for this is that the experiments were conducted with transient transfection, which could lead to variable expression between individual cells. That might affect precise analysis of the effect of the 3KR mutation. To overcome this disadvantage, endogenous genome editing could be an ideal method to knock in the 3KR mutant of caspase-2 at the endogenous locus and compare it to wild type caspase-2, but we feel that this would be beyond the scope of the present manuscript.

Minor concerns

None

Additional non-essential suggestions

I'm curious how the TRAF2-mediated stabilization of the caspase-2 dimer interplays with caspase-2 autocleavage that follows dimerization. Cleavage is considered to stabilize the active enzyme, so does TRAF 2 facilitate this or is it an independent stabilization mechanism. Would TRAF2's ability to stabilize the dimer be altered if autocleavage was blocked by mutating the cleavage sites? This is a little outside the scope of the initial study but inclusion of this in the discussion may be warranted.

The referee raises an intriguing question, which we sought to explore further in the discussion, per the referee's recommendation.

Referee #3:

In this manuscript the authors aim to unveil a role for ubiquitination in the activation of caspase-2. Despite being cloned as one of the first mammalian caspases, the molecular machinery driving the activation of caspase-2 remains unclear. Thus the manuscript addresses a pertinent issue and therefore the story on caspase-2 biology is of potential interest to a broad readership. However, the manuscript suffers from serious technical concerns, over relying on "overexpression and/or nearly endogenous expression" experiments including fragmented caspase-2 coupled to tags. The fact that TRAF2-caspase-2 interaction has been demonstrated before (in the context of NfκB activation and the authors didn't address if these results are reproducible in their settings and if it has any role here) diminishes the enthusiasm to support publication the publication of this manuscript in EMBOJ. Thus the story might be appropriate for a specialized journal if the authors could convincingly address the following concerns.

1) One of the major concerns is the lack of convincing data demonstrating the ubiquitination of caspase-2 at endogenous levels. Most of the experiments are performed with "nearly endogenous" expression of BIFC-CARD domain (its claimed as caspase-2 in the manuscript in many places) in tumour cell lines. I'm surprised that the authors have not attempted well established techniques like "in situ trapping of caspases" with biotin-VAD which has been very well shown by several labs to precipitate the proximal caspases including caspase-2 (under both PIDD dependent and independent conditions). Ideally the authors should precipitate the dimerized caspase-2 with Biotin-VAD upon appropriate stimuli and check for ubiquitination at endogenous levels.

We agree that detecting ubiquitylation on endogenous caspase-2 would help confirm our other work. Therefore, we examined the ubiquitylation of endogenous caspase-2 after cisplatin treatment by denaturing/renaturing immunoprecipitation with anti-caspase-2 antibody, as discussed in response to the referee #1's comment. Please see the result in Figure 5B.

As the referee noted, other groups have successfully been able to precipitate active caspase-2 with biotin-VAD-fmk with heat shock or α -toxin for example (Tu *et al*, 2006; Imre *et al*, 2012). We also attempted to utilize biotin-VAD-fmk, but found it to be ineffective at precipitating dimerized caspase-2 after cisplatin treatment. Although it can be a useful tool, there are several reasons why biotin-VAD-fmk might not be effective at capturing cisplatin-activated caspase-2 in our setting. Stimuli known to promote good biotin-VAD-fmk capture of caspase-2, such as heat shock, induce a fairly rapid and synchronous activation of caspase-2 and cell death (Bouchier-Hayes *et al*, 2009; Imre *et al*, 2012; McStay & Green, 2014; Tu *et al*, 2006). However, DNA damage can induce much more varied responses in a cell population, leading to uneven activation of caspases. Furthermore, previous studies have shown that VAD-based inhibitors bind poorly to caspase-2, in comparison with other caspases (Ekert *et al*, 1999). Based on these observations, as well as the facts that caspase-2 dimerization precedes proteolytic activity and caspase-2 BiFC is currently considered a standard method to monitor caspase-2 dimerization (Bouchier-Hayes *et al*, 2009; Parsons *et al*, 2015; Parsons & Bouchier-Hayes, 2015), we believe that caspase-2 BiFC coupled with GFP-Trap is a more sensitive tool for isolating and studying active caspase-2.

2) The authors failed to convincingly demonstrate that dimerized caspases-2 at endogenous levels is indeed ubiquitinated in a TRAF2 dependent manner. A direct role for ubiquitination in the dimerization /stabilisation of full-length caspase-2 is lacking. Gel filtration/ cross linking experiments could perhaps be employed (and gel filtration has been routinely employed by groups working on caspases-2 before to demonstrate caspases-2 oligomerisation in response to apoptotic stimuli) to demonstrate if dimerized caspase-2 in response to activating stimuli is indeed ubiquitinated in a Traf2 dependent manner. One should attempt the endogenous gel filtration by growing and lysing cells at 37°C and not by shifting the cell lysates from 4°C to 37°C (where caspases-2 can artificially get into high order multimers in a NaCl /KCl dependent manner).

We appreciate the referee's interest in examining endogenous caspase-2 dimers. Unfortunately, we are not aware of any technique that would

allow us to robustly isolate and study endogenous caspase-2 dimers. The lack of such a tool was what helped initiate this project. Regardless, it would be technically difficult to specifically isolate caspase-2 dimers and show that they are ubiquitylated in a TRAF2 dependent manner, since loss of TRAF2 prevents dimerization. We were able to detect ubiquitylation of endogenous caspase-2 by denaturing/renaturing IP, and found that ubiquitylation increased after cisplatin treatment (Figure 5B). However, the ubiquitylation signal was weaker than that obtained by caspase-2 BiFC and GFP-Trap-mediated capture and did not provide an optimal dynamic range for evaluating the effect of TRAF2 knockdown. This is likely because non-dimerized caspase-2 was also precipitated during the experiment and obscured the relevant signal.

GFP-Trap's high affinity and selective capture of dimerized BiFC enables monitoring of the ubiquitylation of the caspase-2 dimer, while antibody-based endogenous caspase-2 immunoprecipitation was much less effective. Therefore, it would be technically prohibitive to detect endogenous caspase-2 ubiquitylation by gel filtration and immunoprecipitation. We instead employed a crosslinking reagent to examine caspase-2 oligomerization. In control cells, caspase-2 oligomerization was increased by cisplatin treatment. In contrast, cisplatin-induced caspase-2 oligomerization was decreased in shTRAF2 cells, consistent with TRAF2-mediated caspase-2 ubiquitylation and oligomerization. The result is in Appendix Figure S3C.

3) TRAF2 /1 has been shown to be interacting with caspases-2 already and this reference is cited only in the discussion. But it should be cited already in page 4, when the authors make the claim that "we discover that caspases-2 dimers interact with TRAFs".

We appreciate the referee's comment. Our intention was to say that we are the first to determine that TRAF2 is a specific interactor of caspase-2 *dimers*, and to implicate TRAF2 in caspase-2 enzymatic activation, providing important findings not previously observed by Lamkanfi et al. (Lamkanfi *et al*, 2005). However, to remove any ambiguity, we changed the wording of the sentence and included a citation earlier in the manuscript.

As TRAF2 knock out cells are readily available the authors could test if caspase-2 activation is impaired in these cells at endogenous levels and if caspase-2 ub'n is impaired in response to cell death stimuli that activates caspase-2. Further, one can check if these cells are resistant to cisplatin and if this is dependent on caspase-2 activation. Again here one can test for dimerization of caspases-2 (biotin-vad)

followed by processing or by cleavage of VDVAD-fmk. Here again, caspase-2 knock out cells could serve as a control to confirm the specificity of VDVAD-fmk cleavage.

It has been reported that caspase-2 functions to suppress tumorigenesis, and that its role could be influenced by genetic background and is more apparent in transformed or cancer cells than in normal cells (Sidi *et al*, 2008; Ho *et al*, 2009; Puccini *et al*, 2013). We therefore generated TRAF2 knockout cells in HeLa by CRISPR/Cas9 technology with lentiviral transduction. Because of the efficiency of the knockout, we performed initial experiments without clonal selection (Appendix Figure S3F). Surprisingly, TRAF2 CRISPR-KO HeLa cells showed similar levels of cisplatin-induced caspase-2 activation, as monitored by caspase-2 BiFC (Appendix Figure S3D), and cell death compared to control cells (Appendix Figure S3E). We were confident in our RNAi experiments, as we had used several different sequences for TRAF2 knockdown by siRNA and shRNA, and all results consistently supported that TRAF2 is necessary for caspase-2 activation and apoptosis in response to cisplatin. Recently it was reported that gene disruption, like CRISPR/Cas9-mediated gene knockout, can cause genetic compensation by upregulation of other gene(s) in the same signaling pathway (Rossi *et al*, 2015). In that report, genetic compensation was induced by gene knockout (by TALEN-based gene disruption), but not by post-transcriptional gene knockdown (by RNAi or CRISPR interference). Indeed, we found that TRAF3 expression was increased in TRAF2 knockout cells (Appendix Figure S3F). This upregulation was even more pronounced in single colony-derived clones (Appendix Figure S3G). However, such TRAF3 upregulation was not observed in RNAi-mediated TRAF2 knockdown (Figure 3B and Appendix Figure S3F). Although TRAF3 was not as critical for caspase-2 activation as TRAF2, as demonstrated by the siRNA-mediated knockdown experiment in Casp2pro BiFC cells (Figure 3B), an upregulation of TRAF3 could compensate for the permanent loss of TRAF2 when TRAF2 is genetically disrupted by CRISPR/Cas9. Supporting this hypothesis, siRNA-mediated knockdown of TRAF3 in TRAF2 knockout cells significantly reduced caspase-2 dimerization (Appendix Figure S3H). Importantly, TRAF3 knockdown did not have a significant effect on caspase-2 dimerization in control cells, indicating that loss of TRAF2 induces a compensatory upregulation of TRAF3, which then becomes the primary activator of caspase-2 in this context.

Unfortunately, examining initiator caspase activity or cleavage following biotin-VAD-fmk capture is not possible as the inhibitor covalently binds the caspase active cysteine site irreversibly (Tu *et al*, 2006). Even if we

could use biotin-VAD-fmk to monitor caspase-2 activation, a caspase-2 processing or VDVAD cleavage assay would not be feasible after biotin-VAD-fmk precipitation.

4) The data on ubiquitination is not robust. To test *in vivo* ubiquitination of endogenous caspase-2 one should employ a denaturing/renaturing IP. Else TUBEs could also be employed. Ideally, one should precipitate Ub'nd proteins upon cisplatin stimulation and check for caspases-2 in TRAF2 knock out cells and in TRAF2 knock out cells complimented with wild type and RING deficient mutants. The authors can check for caspase-2 processing and cell death induction in the same assays. This will also infer if the caspases-2/TRAF-2 required for mediating cisplatin-mediated cell death in a cell type dependent manner.

Unfortunately, as we mentioned in reply to the previous comment, TRAF2 knockout cells were unsuitable for such experiments because of genetic compensation by TRAF3 upregulation. However, we were able to validate that cisplatin induces an interaction between endogenous caspase-2 and endogenous TRAF2 (Figure 2G), and that cisplatin induces ubiquitylation of endogenous caspase-2 (Figure 5B). At the same time, the detection of endogenous caspase-2 ubiquitylation was much less effective than that by Casp2-BiFC and GFP-Trap, and it was difficult see a robust reduction by TRAF2 knockdown. Please also refer to our response to comment #2.

5) It is surprising that the authors employed primarily the CARD -BIFC for most of the experiments and that's probably the reason why most of the Ubiquitination sites are identified in the N-terminus. These experiments should at the least be complimented with full-length catalytically dead caspase-2 mutants. Ideally, the authors should employ a Lys-Gly-GLY approach or other mass spec based approaches in an unbiased manner and look for ubiquitination sites on the entire caspase-2 protein precipitated from cells upon stimulation with DNA damaging agents. Or this can also be attempted on recombinant caspases-2 subjected to ubiquitination with TRAF-2. The identified sites could then be verified *in vivo*.

We primarily utilized the CARD-containing caspase-2 prodomain for our initial BiFC experiments for several reasons: (i) initiator caspase prodomains are generally considered to be the docking sites for important regulators, like activation platform components (Shi, 2002; Boatright *et al*, 2003; Kumar, 1999; Duan & Dixit, 1997; Bouchier-Hayes & Green, 2012; Park *et al*, 2007); (ii) Drs. Bouchier-Hayes and Green established the validity of using the caspase-2 prodomain for BiFC when they first described the technique (Bouchier-Hayes *et al*, 2009), and it has since been used by many other groups; and (iii) we were unable to create cell lines with stable and functional integration of the bidirectional

expression cassette containing full length Casp2 BiFC, most likely because the large size of the construct made proper integration difficult.

Using the caspase-2 prodomain we were able to identify TRAF protein interactions and demonstrate the necessity of those interactions for dimerization. It thus seemed reasonable that relevant TRAF2-mediated ubiquitylation sites would also be located in the prodomain, which is why we initially focused there. However, we have utilized full length caspase-2 to validate the effect of the 3KR mutation on caspase-2 ubiquitylation (Figures 5D and 5E), TRAF2 binding (Figures 5H and 5J), localization (Figures 6A, 6B, 6D, and Appendix Figure S6B), dimerization (Figures 7A and 7B), and activity (Figures 7D-G, and Appendix Figure S7A).

Nevertheless, we are interested in the identification of ubiquitylation sites outside of prodomain, and how those ubiquitylation events may affect caspase-2. However, in our opinion that is beyond the scope of the present manuscript, which focuses on the regulation of caspase-2 dimerization and activation.

6) The requirement of caspase-2 in DNA damage-mediated apoptosis is controversial and may be cell type dependent. While the authors tend to claim that they are looking at a scenario where caspases-2 activation is independent of PIDD and RAIDD, further experiments employing loss of function studies are needed to substantiate their claims here.

The other referees suggested this as well, so we made sure to examine the involvement of RAIDD and PIDD. Please see Appendix Figure S2B-F. Based on these results, caspase-2 activation seems to be independent of RAIDD and PIDD, at least in response to cisplatin treatment in HeLa cells.

Along these lines, it is not clear if caspases-3/7 is required for the activation of caspases-2 under these settings. IF we go with published literature, one may need effector caspases for activation and /or processing of caspase-2 which also questions the role for Ub'n here. It is indeed possible that caspases-2 is activated in the absence of caspases-3/7 in their settings as a proximal caspase - But this needs to be demonstrated with employing caspases-3/7 double knock out cells.

The literature mentioned by the referee could be a study from Dr. Villunger's group (Manzl *et al*, 2009) where they showed that caspase-2 can be cleaved downstream of caspases-3/7 activation. However, as the protocol developed by Dr. Bouchier-Hayes recommended, our caspase-2 BiFC experiments were conducted in the presence of the pan-caspase inhibitor Q-VD(OMe)-OPh to avoid effector caspase activation (Bouchier-

Hayes *et al*, 2009; Parsons & Bouchier-Hayes, 2015). We also added Q-VD(OMe)-OPh to cell culture when endogenous caspase-2 ubiquitylation by cisplatin was examined. Therefore, we believe that caspases-3/7 should not be involved in caspase-2 dimerization/ubiquitylation.

7) Does TRAF2 overexpression stimulate caspases-2 activation in a RING dependent manner?

As we showed in Figure 5D, TRAF2 overexpression induced caspase-2-activating ubiquitylation in a manner dependent on the RING domain of TRAF2. In addition to that, we examined whether TRAF2 overexpression affects cell death. Consistent with TRAF2-induced caspase-2 ubiquitylation, TRAF2 overexpression induced apoptotic cell death, but the TRAF2 RING domain mutant was far less potent. Cell death induced by wild-type TRAF2 was decreased in caspase-2 knockdown cells, further supporting the biological significance of TRAF2-mediated caspase-2 activation. The results are in Appendix Figures S7B and S7C.

8) What kind of ubiquitin chains is synthesized on caspase-2 by TRAF2? Again here, mass spec/TUBE based approaches could shed some light which will enhance the mechanistic insights provided.

Please see our reply to referee #1 and Appendix Figure S5D, as referee #1 had a similar concern.

9) It's also not clear what is the functional significance of the translocation/shift of caspase-2 to detergent insoluble fraction. Again in Figure 6E, there is no evidence presented to demonstrate if the endogenous caspases-2 is ubiquitinated in the insoluble fraction.

Our result indicates that ubiquitylation of caspase-2 enhances translocation and further activation since the 3KR mutation reduced both. The importance of the insoluble fraction is not clear, but it is possible that this fraction contains an unknown factor that is required for full caspase-2 activation, or perhaps an apoptosis-promoting substrate targeted by caspase-2. We also examined endogenous caspase-2 ubiquitylation in the detergent-insoluble fraction after cisplatin treatment. The result is in Appendix Figure S6C. As in Figure 5B where we showed endogenous caspase-2 ubiquitylation in whole cell lysate, endogenous caspase-2 ubiquitylation was observed in both the soluble and the insoluble fractions. While the ubiquitylation signal in the insoluble fraction was just slightly higher than that in the soluble fraction, the proportion of ubiquitylated to non-ubiquitylated caspase-2 is much higher in the insoluble fraction indicating that ubiquitylated caspase-2 is enriched in

this fraction.

10) In Figure 2, there is no evidence presented to claim that the interaction between TRAF2 and caspases-2 is induced upon cisplatin stimulation at endogenous levels.

Please see our response to the first comment from referee #2 and Figure 2G.

11) In figure 3F: The authors should check for caspases-2 processing. Though cleavage is a secondary event and not necessarily an indication of activation of proximal caspases like caspases-2, a block in processing of caspase-2 could possibly support an upstream obligatory role for TRAF2 in caspases-2 activation. Further, is caspase-9 activation is also blocked?

We checked the cleavage of caspase-2 and caspase-9 in shTRAF2 cells. Consistent with TRAF2 being necessary for caspase-2 activation, cleavage of both caspase-2 and caspase-9 were suppressed in shTRAF2 cells. The result is in Figure 3F.

12) As TRAF-2/Caspase-2 complex is shown to regulate NfKB, there is no evidence presented to argue that this pathway is not activated in response to the stimuli employed.

We first tested whether the NF- κ B pathway is activated by caspase-2 overexpression, as reported by Lamkanfi *et al.* (Lamkanfi *et al.*, 2005). Consistent with the previous report, caspase-2 overexpression induced canonical NF- κ B pathway activation, which was confirmed by phosphorylation of p65 and I κ B α , and conversion of NF- κ B1 p105 to p50 (Appendix Figure S2H). Cisplatin treatment also induced NF- κ B pathway activation, although not as strongly as caspase-2 overexpression or treatment with TNF α , a well-known NF- κ B pathway activator (Appendix Figures S2H and S2I). Importantly, despite the fact that cisplatin somewhat activated the NF- κ B pathway, NF- κ B pathway inhibitors, TPCA-1 and IKK-16, failed to suppress cisplatin-induced caspase-2 BiFC. This implies that the NF- κ B pathway is not involved in caspase-2 dimerization and activation in response to cisplatin treatment (Appendix Figure S2J).

We should add that the NF- κ B pathway is generally considered to be a pro-survival pathway – indeed, previous studies have shown that this pathway promotes resistance to DNA damage-inducing chemotherapy (Bottero *et al.*, 2001). Therefore, we consider it unlikely that caspase-2 would induce apoptosis through the NF- κ B pathway in response to

cisplatin. Indeed, we have observed that inhibiting the NF- κ B pathway can actually enhance cisplatin-induced cell death (ACR, personal observations).

13) Figure 4: Are the TAQEM mutations impair the activation and processing of caspase-2 directly?

We cannot exclude the possibility that the mutation impairs the catalytic activity of caspase-2 as the referee suggested. Nevertheless, there is some evidence that mutating the TRAF interacting motif did not strongly disrupt the structure of caspase-2. The TIM mutant decreased both the interaction of the TRAFs with caspase-2 (Figure 4B) and caspase-2 dimerization (Figure 4D). However, it could still bind RAIDD as well as wild type caspase-2. These results suggest that the TIM mutation likely did not negatively alter the protein structure of caspase-2. The result is in Appendix Figure S4A.

14) Figure 6A: The "puncta" needs to be quantified from multiple fields and from multiple experiments. As of now this figure is not so informative.

Per the referee's comment, caspase-2 punctate signals were quantified from at least 3 different fields of 4 biological replicate experiments. The percentage of the punctate signals were specified in Figure 6A.

15) Figure 6D: Though overexpressed/ expressed to near endogenous levels, is the caspases-2(C320A)-venus ubiquitinated in the insoluble fraction?

We examined the ubiquitylation of caspase-2(C320A)-mVenus expressed in HeLa cells. Cells were treated with cisplatin and fractionated for detergent-soluble or -insoluble fraction. In response to cisplatin treatment, ubiquitylation of caspase-2(C320A)-mVenus was increased more in the insoluble fraction than in the soluble fraction. The result is in Appendix Figure S6B.

REFERENCES

- Baptiste-Okoh N, Barsotti AM & Prives C (2008) A role for caspase 2 and PIDD in the process of p53-mediated apoptosis. *Proc Natl Acad Sci USA* **105**: 1937–1942
- Boatright KM, Renatus M, Scott FL, Sperandio S, Shin H, Pedersen IM, Ricci J-E, Edris WA, Sutherlin DP, Green DR & Salvesen GS (2003) A unified model for apical caspase activation. *Mol Cell* **11**: 529–541
- Bottero V, Busuttill V, Loubat A, Magné N, Fischel JL, Milano G & Peyron JF (2001) Activation of nuclear factor kappaB through the IKK complex by the topoisomerase poisons SN38 and doxorubicin: a brake to apoptosis in HeLa human carcinoma cells. *Cancer Res* **61**: 7785–7791
- Bouchier-Hayes L & Green DR (2012) Caspase-2: the orphan caspase. *Cell Death Differ* **19**: 51–57
- Bouchier-Hayes L, Oberst A, McStay GP, Connell S, Tait SWG, Dillon CP, Flanagan JM, Beere HM & Green DR (2009) Characterization of cytoplasmic caspase-2 activation by induced proximity. *Mol Cell* **35**: 830–840
- Duan H & Dixit V (1997) RAIDD is a new ‘death’ adaptor molecule. *Nature* **385**: 86–89
- Ekert PG, Silke J & Vaux DL (1999) Caspase inhibitors. *Cell Death Differ* **6**: 1081–1086
- Fava LL, Schuler F, Sladky V, Haschka MD, Soratroi C, Eiterer L, Demetz E, Weiss G, Geley S, Nigg EA & Villunger A (2017) The PIDDosome activates p53 in response to supernumerary centrosomes. *Genes Dev* **31**: 34–45
- Ho LH, Taylor R, Dorstyn L, Cakouros D, Bouillet P & Kumar S (2009) A tumor suppressor function for caspase-2. *Proc Natl Acad Sci USA* **106**: 5336–5341
- Imre G, Heering J, Takeda A-N, Husmann M, Thiede B, Zu Heringdorf DM, Green DR, van der Goot FG, Sinha B, Dötsch V & Rajalingam K (2012) Caspase-2 is an initiator caspase responsible for pore-forming toxin-mediated apoptosis. *EMBO J* **31**: 2615–2628
- Kumar S (1999) Mechanisms mediating caspase activation in cell death. *Cell Death Differ* **6**: 1060–1066
- Lamkanfi M, D'hondt K, Vande Walle L, van Gurp M, Denecker G, Demeulemeester J, Kalai M, Declercq W, Saelens X & Vandenamele P (2005) A novel caspase-2 complex containing TRAF2 and RIP1. *J Biol Chem* **280**: 6923–6932

- Lin Y, Ma W & Benchimol S (2000) Pidd, a new death-domain-containing protein, is induced by p53 and promotes apoptosis. *Nat Genet* **26**: 122–127
- Manzl C, Krumschnabel G, Bock F, Sohm B, Labi V, Baumgartner F, Logette E, Tschopp J & Villunger A (2009) Caspase-2 activation in the absence of PIDDosome formation. *J Cell Biol* **185**: 291–303
- McStay GP & Green DR (2014) Identification of active caspases using affinity-based probes. *Cold Spring Harb Protoc* **2014**: pdb.prot080309
- Ohtake F, Saeki Y, Ishido S, Kanno J & Tanaka K (2016) The K48-K63 Branched Ubiquitin Chain Regulates NF- κ B Signaling. *Mol Cell* **64**: 251–266
- Oliver TG, Meylan E, Chang GP, Xue W, Burke JR, Humpton TJ, Hubbard D, Bhutkar A & Jacks T (2011) Caspase-2-Mediated Cleavage of Mdm2 Creates a p53-Induced Positive Feedback Loop. *Mol Cell* **43**: 57–71
- Park HH, Logette E, Raunser S, Cuenin S, Walz T, Tschopp J & Wu H (2007) Death domain assembly mechanism revealed by crystal structure of the oligomeric PIDDosome core complex. *Cell* **128**: 533–546
- Parsons MJ & Bouchier-Hayes L (2015) Measuring initiator caspase activation by bimolecular fluorescence complementation. *Cold Spring Harb Protoc* **2015**: pdb.prot082552
- Parsons MJ, Rehm M & Bouchier-Hayes L (2015) Imaging-based methods for assessing caspase activity in single cells. *Cold Spring Harb Protoc* **2015**: pdb.top070342
- Puccini J, Shalini S, Voss AK, Gatei M, Wilson CH, Hiwase DK, Lavin MF, Dorstyn L & Kumar S (2013) Loss of caspase-2 augments lymphomagenesis and enhances genomic instability in Atm-deficient mice. *Proc Natl Acad Sci USA* **110**: 19920–19925
- Rossi A, Kontarakis Z, Gerri C, Nolte H, Hölper S, Krüger M & Stainier DYR (2015) Genetic compensation induced by deleterious mutations but not gene knockdowns. *Nature* **524**: 230–233
- Shi Y (2002) Mechanisms of caspase activation and inhibition during apoptosis. *Mol Cell* **9**: 459–470
- Sidi S, Sanda T, Kennedy RD, Hagen AT, Jette CA, Hoffmans R, Pascual J, Imamura S, Kishi S, Amatruda JF, Kanki JP, Green DR, D'Andrea AA & Look AT (2008) Chk1 suppresses a caspase-2 apoptotic response to DNA damage that bypasses p53, Bcl-2, and caspase-3. *Cell* **133**: 864–877
- Tu S, McStay GP, Boucher L-M, Mak T, Beere HM & Green DR (2006) In situ trapping of activated initiator caspases reveals a role for caspase-2

in heat shock-induced apoptosis. *Nat Cell Biol* **8**: 72–77

Yau RG, Doerner K, Castellanos ER, Haakonsen DL, Werner A, Wang N, Yang XW, Martinez-Martin N, Matsumoto ML, Dixit V & Rape M (2017) Assembly and Function of Heterotypic Ubiquitin Chains in Cell-Cycle and Protein Quality Control. *Cell* **171**: 918–933

Figure R. Ubiquitylation status of caspase-2 has minor effect on RAIDD binding

HEK293T cells were transfected with Casp2(C320A)-mVenus constructs (wild type or K15/152/153R (3KR) mutant) and cultured for 48 h. Then lysates were prepared, followed by GFP-Trap IP. Immunoprecipitates were analyzed by IB for RAIDD binding to Casp2-mVenus.

Thank you for submitting your revised manuscript to The EMBO Journal. Your study has now been re-reviewed by the three referees and the comments are provided below. While Referee #1 is not convinced that the analysis provides enough conclusive data to support the key findings. Referees #2 and 3 are more supportive and find that the current dataset is strong enough as is. I have carefully looked at the concerns raised and also discussed things further with the referees. I do appreciate the introduced revisions and that the data supports the conclusions. I am therefore happy to say that we can offer publication here without further experiments.

REFEREE REPORTS

Referee #1:

In the revised version of the manuscript the authors provide additional evidence that TRAF2-mediated ubiquitination of the Caspase-2 prodomain can promote dimerization-dependent activation of this protease and that ubiquitination increases dimer stability. The authors also show now that caspase-2 can be found ubiquitinated at endogenous levels and that knock-down of TRAF2 or Casp2 can reduce cisplatin-killing of HeLa and BT47 breast cancer cells.

Overall, I think the manuscript has improved but given the time taken for revision, the output is less exciting. There are several limitations of the study that remain.

It seems that neither Casp-2 activation nor cell death depend on PIDD1 or RAIDD, as demonstrated now in siRNA experiments. At least the latter is surprising, given the established role for RAIDD in C2 dimerization and activation and the fact that it copurifies after cisplatin treatment.

All experiments shown actually do suggest that TRAF2 and C2 are involved in cisplatin killing of HeLa cells, but no experiment actually shows the epistatic dependence, i.e. that TRAF2 requires C2 and vice versa. The fact that TRAF2 also binds to Caspase-9 complicates interpretation of the data related to the effects of TRAF2 siRNA on cell death. The fact that TRAF2 KO cells respond normally to cisplatin does not help. Is there more TRAF3 bound to C2 in these cells when compared to their wt counterparts; as suggested by the authors, TRAF3 might compensate here but is initially a rather poor binder.

The cytokinesis issue has been only superficially addressed by showing that AuroraB inhibition also induces BiFC activity, while DHCB seems a poor activator. But it has not been explored if cells do fail cytokinesis at all or that cells that do light up in the context of cisplatin treatment have failed cytokinesis or not (early in the experiment, before they dismantle during apoptosis). Hence, we cannot exclude that apoptosis post cytokinesis failure requires C2 activation in HeLa cells, or if this is really linked to DNA damage induced killing. Same holds true for the BT74 cells.

Still, most of the data relies on overexpression of the pro-domain of C2, or mutant versions of C2 where we do not know if they have normal catalytic activity (e.g. TIM mutant variant) and whether this is causal for the loss of annexin V positive cells in Fig 4

Finally, the shift of a fraction of C2 into insoluble fractions remains very vague and descriptive; what is in this insoluble fraction. Nuclei, nucleoli or centrosomes. Using lamin as a marker suggests nuclei.

As it stands now, I don't think this is an EMBO paper but rather JBC.

Referee #2:

In the revised manuscript by Robeson et al, the authors did a lot of work and added a number of new experiments to address all the reviewers' responses. In particular, I am satisfied that my concerns were suitably addressed by adding experiments showing binding of the endogenous proteins;

binding studies with caspase-2 deletion mutants; and determining if there could be any interference between the overexpressed caspase and the endogenous protein (there is not). Overall, the reviews were extensive and I am impressed that the authors were able to address the majority of the critiques. This has resulted in a stronger study. It is my opinion that this study will make a significant contribution to the general caspase field both in terms of the technical advance presented and the new insights regarding how caspase-2 is activated.

Referee #3:

The authors have performed a substantial set of experiments to address almost all the major concerns raised by us in the initial round of review. A lot of follow up work is suggested but the crucial issues have been addressed successfully and therefore I recommend publication of this manuscript.

2nd Revision - authors' response

30th April 2018

Referee #1:

In the revised version of the manuscript the authors provide additional evidence that TRAF2-mediated ubiquitination of the Caspase-2 prodomain can promote dimerization-dependent activation of this protease and that ubiquitination increases dimer stability. The authors also show now that caspase-2 can be found ubiquitinated at endogenous levels and that knock-down of TRAF2 or Casp2 can reduce cisplatin-killing of HeLa and BT47 breast cancer cells.

Overall, I think the manuscript has improved but given the time taken for revision, the output is less exciting. There are several limitations of the study that remain.

It seems that neither Casp-2 activation nor cell death depend on PIDD1 or RAIDD, as demonstrated now in siRNA experiments. At least the latter is surprising, given the established role for RAIDD in C2 dimerization and activation and the fact that it copurifies after cisplatin treatment.

We were also surprised to find that RAIDD was involved in neither caspase-2 dimerization nor apoptosis following DNA damage using the cell types and damaging agents we employed. However, other groups have also found RAIDD to be dispensable for caspase-2 activation (Manzl *et al*, 2009; Imre *et al*, 2012; Peintner *et al*, 2015). It is unclear what function RAIDD binding of caspase-2 plays in this context; while it would be interesting to explore this further, we felt it was beyond the scope of this manuscript.

All experiments shown actually do suggest that TRAF2 and C2 are involved in cisplatin killing of HeLa cells, but no experiment actually shows the epistatic dependence, i.e. that TRAF2 requires C2 and vice versa. The fact that TRAF2 also binds to Caspase-9 complicates interpretation of the data related to the effects of

TRAF2 siRNA on cell death. The fact that TRAF2 KO cells respond normally to cisplatin does not help. Is there more TRAF3 bound to C2 in these cells when compared to their wt counterparts; as suggested by the authors, TRAF3 might compensate here but is initially a rather poor binder.

It is true that we did not show formally that TRAF2 requires caspase-2 to induce cell death in response to DNA damage. However, we did show that, in response to DNA damage, caspase-2 was dependent on TRAF2 for ubiquitylation and dimerization, which facilitates the full activation of caspase-2.

We do not know the precise effect of TRAF2 binding to caspase-9 (it could be pro- or anti-apoptotic). We believe that this is of interest with respect to TRAF2 biology. However, we did see that TRAF2 knockdown had an effect on cell death upstream of mitochondrial outer membrane permeabilization, indicating that the diminished cell death was not likely to result from the lack of a direct interaction between TRAF2 and caspase-9.

As we showed in Figures EV3F, EV3G, and EV3H, TRAF3 is upregulated in the absence of TRAF2, and in this context becomes critical for caspase-2 activation. We are not sure if this means that, in this context, TRAF3 interacts more strongly with caspase-2; this merits future exploration.

The cytokinesis issue has been only superficially addressed by showing that AuroraB inhibition also induces BiFC activity, while DHCB seems a poor activator. But it has not been explored if cells do fail cytokinesis at all or that cells that do light up in the context of cisplatin treatment have failed cytokinesis or not (early in the experiment, before they dismantle during apoptosis). Hence, we cannot exclude that apoptosis post cytokinesis failure requires C2 activation in HeLa cells, or if this is really linked to DNA damage induced killing. Same holds true for the BT74 cells.

While we have not entirely ruled out the possibility that a failure of cytokinesis is responsible for caspase-2 activation in response to DNA damage, we think it is unlikely. The important work of Fava et al. (Fava et al, 2017), demonstrated that both PIDD and RAIDD were necessary for caspase-2 activation in response to a failure of cytokinesis, while we found that the PIDDosome was dispensable. We would also point out that Fava et al. used cleavage of MDM2 (a known caspase-2 substrate) as their readout for caspase-2 activation instead of dimerization or Bid cleavage. This could have limited their ability to detect caspase-2 activation in response to DNA damage. Based on our work and the work of others, we speculate that the function and regulation of caspase-2 depends on multiple factors, such as the stage of the cell cycle or the

absence of p53, and further work is certainly needed to address this.

Still, most of the data relies on overexpression of the pro-domain of C2, or mutant versions of C2 where we do not know if they have normal catalytic activity (e.g. TIM mutant variant) and whether this is causal for the loss of annexin V positive cells in Fig 4

We understand the challenge of studying caspase-2 activity, given that overexpression can unintentionally induce dimerization. That is why we wanted to create a caspase-2 system that expressed at near endogenous levels, which we believe we were able to do. With regard to the TIM mutant, it is true that we cannot rule out the possibility that the mutation impairs the catalytic activity of caspase-2 directly. However, we did find that the TIM mutant showed a moderate ability to dimerize (Figure 4D) and was still able to bind RAIDD as well as wild type caspase-2 (Figure EV4A), indicating it maintained some proper structure. Please see our response to the comment #13 of referee #3 in our previous point-by-point response.

Finally, the shift of a fraction of C2 into insoluble fractions remains very vague and descriptive; what is in this insoluble fraction. Nuclei, nucleoli or centrosomes. Using lamin as a marker suggests nuclei.

We have assessed what is included in the detergent-insoluble fraction by the expression of tagged organelle markers. The result is shown in Figure R2 for the reviewer's consideration (please see the end of this letter). It appears that chromatin, golgi, and peroxisomes are all included in the insoluble fraction. We have not tested specifically for nucleolar or centrosomal markers, but it is reasonable to speculate that these organelles are included in the insoluble fraction, which would be in line with recent reports (Ando *et al*, 2017; Fava *et al*, 2017).

As it stands now, I dont think this is an EMBO paper but rather JBC.

Referee #2:

In the revised manuscript by Robeson et al, the authors did a lot of work and added a number of new experiments to address all the reviewers' responses. In particular, I am satisfied that my concerns were suitably addressed by adding experiments showing binding of the endogenous proteins; binding studies with caspase-2 deletion mutants; and determining if there could be any interference between the overexpressed caspase and the endogenous protein (there is not). Overall, the reviews were extensive and I am impressed that the authors were able to address

the majority of the critiques. This has resulted in a stronger study. It is my opinion that this study will make a significant contribution to the general caspase field both in terms of the technical advance presented and the new insights regarding how caspase-2 is activated.

Referee #3:

The authors have performed a substantial set of experiments to address almost all the major concerns raised by us in the initial round of review. A lot of follow up work is suggested but the crucial issues have been addressed successfully and therefore I recommend publication of this manuscript.

REFERENCES

- Ando K, Parsons MJ, Shah RB, Charendoff CI, Paris SL, Liu PH, Fassio SR, Rohrman BA, Thompson R, Oberst A, Sidi S & Bouchier-Hayes L (2017) NPM1 directs PIDDosome-dependent caspase-2 activation in the nucleolus. *J Cell Biol* **216**: 1795–1810
- Fava LL, Schuler F, Sladky V, Haschka MD, Soratroi C, Eiterer L, Demetz E, Weiss G, Geley S, Nigg EA & Villunger A (2017) The PIDDosome activates p53 in response to supernumerary centrosomes. *Genes Dev* **31**: 34–45
- Imre G, Heering J, Takeda A-N, Husmann M, Thiede B, Zu Heringdorf DM, Green DR, van der Goot FG, Sinha B, Dötsch V & Rajalingam K (2012) Caspase-2 is an initiator caspase responsible for pore-forming toxin-mediated apoptosis. *EMBO J* **31**: 2615–2628
- Manzl C, Krumschnabel G, Bock F, Sohm B, Labi V, Baumgartner F, Logette E, Tschopp J & Villunger A (2009) Caspase-2 activation in the absence of PIDDosome formation. *J Cell Biol* **185**: 291–303
- Peintner L, Dorstyn L, Kumar S, Aneichyk T, Villunger A & Manzl C (2015) The tumor-modulatory effects of Caspase-2 and Pidd1 do not require the scaffold protein Raidd. *Cell Death Differ* **22**: 1803–1811

Figure R2. Organelle included in detergent-soluble and -insoluble fractions
HeLa cells were transfected with plasmids encoding the indicated organelle markers with mCherry tags. Cells were biochemically fractionated into detergent-soluble (S) and -insoluble (I) fractions, and analyzed by IB with anti-mCherry antibody.

Plasmids encoding mCherry-tagged organelle markers were gifts from Dr. Michael Davidson. Addgene plasmid names and numbers are as follows:

cytoskeleton: mCherry-Tubulin-6, #55147

nucleus (chromatin): mCherry-H2A-10, #55054

mitochondria: mCherry-Mito-7, #55102

golgi: mCherry-Golgi-7, #55052

peroxisome: mCherry-PMP-N-10, #55120

endoplasmic reticulum: mCherry-ER-3, #55041

autophagosome: mCherry-Sequestosome1-C-18, #55131

Corresponding Author Name: Sally Kornbluth and Kenryo Matsuura

Journal Submitted to: The EMBO Journal

Manuscript Number: EMBOJ-2017-97072